# Prototypical Self-Explainable Models Without Re-training

**Srishti Gautam**[*]                                                                 *srishti.gautam@uit.no*
*Department of Physics and Technology*
*UiT The Arctic University of Norway, Norway*

**Ahcene Boubekki**[*]                                                               *ahcene.boubekki@ptb.de*
*Machine Learning and Uncertainty*
*Physikalisch-Technische Bundesanstalt, Germany*

**Marina M. C. Höhne**                                                         *marina.hoehne@uni-potsdam.de*
*Data Science in Bioeconomy, Leibniz Institute for Agriculture and Bioeconomy*
*Institute for Computer Science, University of Potsdam, Germany*

**Michael C. Kampffmeyer**                                                     *michael.c.kampffmeyer@uit.no*
*Department of Physics and Technology*
*UiT The Arctic University of Norway, Norway*

**Reviewed on OpenReview:** *https://openreview.net/forum?id=HU5DOUp6Sa*

## Abstract

Explainable AI (XAI) has unfolded in two distinct research directions with, on the one hand, post-hoc methods that explain the predictions of a pre-trained black-box model and, on the other hand, self-explainable models (SEMs) which are trained directly to provide explanations alongside their predictions. While the latter is preferred in safety-critical scenarios, post-hoc approaches have received the majority of attention until now, owing to their simplicity and ability to explain base models without retraining. Current SEMs, instead, require complex architectures and heavily regularized loss functions, thus necessitating specific and costly training. To address this shortcoming and facilitate wider use of SEMs, we propose a simple yet efficient universal method called KMEx (K-Means Explainer), which can convert any existing pre-trained model into a prototypical SEM. The motivation behind KMEx is to enhance transparency in deep learning-based decision-making via class-prototype-based explanations that are diverse and trustworthy without retraining the base model. We compare models obtained from KMEx to state-of-the-art SEMs using an extensive qualitative evaluation to highlight the strengths and weaknesses of each model, further paving the way toward a more reliable and objective evaluation of SEMs[1].

## 1 Introduction

XAI has become a key research area with the primary objective of enhancing the reliability of deep learning models (Yosinski et al., 2015; Tjoa & Guan, 2021). This domain has notably evolved along two parallel trajectories in recent years. One focuses on post-hoc methods (Ribeiro et al., 2016; Selvaraju et al., 2017), where the algorithms aim to explain the behavior of the black-box models *after* they have been trained. The other promising branch focuses on SEMs (Rudin, 2019), where the models are strategically designed and trained to generate explanations *along with* their predictions.

The easily employable post-hoc techniques have become widely adopted in recent works due to their ability to offer insights into any black-box models without retraining (Bodria et al., 2023). Nevertheless, the need

---

[*]Equal contribution
[1]The code is available at https://github.com/SrishtiGautam/KMEx

for inherently interpretable models has taken some momentum fueled by the unreliability and high variability of these post-hoc methods, which inhibits their usability for safety-critical applications (Rudin, 2019). SEMs offer explanations that align with the actual computations of the model, thus proving to be more dependable, which is crucial in domains such as criminal justice, healthcare, and finance (Rudin, 2019). However, existing SEMs rely on complex designs based on large deep-learning backbones and require intricate training strategies. The associated computational and time costs limit their accessibility and sustainability.

We tackle this limitation by introducing a simple but efficient method called KMEx (K-Means Explainer), which is the first approach that aims to convert a trained black-box model into a prototypical self-explainable model (PSEM). PSEMs provide inherent explanations in the form of class-representative concepts, also called prototypes, in the latent space that can be visualized in the human-understandable input-space (Kim et al., 2021). These prototypes serve as global explanations of the model (*this looks like that* (Chen et al., 2019b)), and their visualization provides knowledge about their neighborhood in the learned embedding. KMEx keeps the trained encoder intact, learns prototypes via clustering in the embedding space, and replaces the classifier with a transparent one. This results in an SEM with similar local explanations and performance to the original black-box model, and such enables the reuse of existing trained models.

Comparing models obtained using KMEx to existing PSEMs requires a comprehensive evaluation strategy which, for this fairly new field, is still lacking. Differing from conventional black-box classifiers, PSEMs yield global (prototypes' visualization) and/or local (activation of individual prototypes by input images) explanation maps alongside the predicted class probabilities. Yet, the assessment of SEMs until now has been limited to comparing the predictive performance to the black-box counterpart with the same backbone architecture as the SEM, followed by quantifying the robustness of local explanation maps and qualitative evaluation of global explanations Wang & Wang (2021); Parekh et al. (2021). We argue that this approach overlooks crucial facets of SEM explainability, failing to establish a standardized framework for thorough analysis and comparison of existing models. For example, we observe that most of the prototypes learned by recent SEMs might never be used by their classifier, which challenges the rationale of a transparent model. Further, the diversity captured by different prototypes in the embedding space, while being a driving force behind the development of several SEMs (Wang et al., 2021), has traditionally only been validated by highly subjective visual inspection of the prototypes.

We, therefore, present a novel quantitative and objective evaluation framework based on the three properties that arose as predicates for SEMs (Gautam et al., 2022): transparency, diversity, and trustworthiness. The rationale is not to rank models but to highlight the consequences of modeling choices. Indeed, in some applications, having robust local explanations might be more valuable than diverse prototypes. Yet, this behavior needs to be quantified in order to support practitioners in choosing the best model for their use case.

Our main contributions are thus as follows:

- We propose a simple yet efficient method, KMEx, which converts any existing black-box model into a PSEM, thus enabling wider applicability of SEMs.
- We propose a novel quantitative evaluation framework for PSEMs, grounded in the validation of SEM's predicates (Gautam et al., 2022), which allows for an objective and comprehensive comparison. Nonetheless, it is important to note that the interpretability of the model still relies on the visual analysis of the prototypes as well as their similarity with test data.

Our key findings are as follows:

- Experiments on various datasets confirm that KMEx matches the performance of the black-box model while offering inherent interpretability without altering the embedding, making it an efficient benchmark for SEMs.
- Most existing PSEMs tend to *ghost* the prototypes, i.e., never utilize them for prediction, which gives a false sense of needed concepts but also undermines the rationale formalized by the predicates, especially transparency.
- Unlike KMEx, the large variations in the design and regularizations of other SEMs lead to drastically different learned representation spaces and local explanations.
- While many SEMs incorporate measures to obtain diverse prototypes, these efforts are not necessarily reflected in terms of captured input data attributes. We illustrate how KMEx can be leveraged, without

Table 1: Design strategies used by state-of-the-art SEMs.

|  | Similarity Measure | Classifier | Prototypes | Diversity Loss |
|---|---|---|---|---|
| ProtoPNet | Distance based | Linear Layer | Projected from training data | Min/max intra/inter-class distance |
| FLINT | Linear Layer | Linear Layer | Weight of the network | Min/max similarity entropy |
| ProtoVAE | Distance based | Linear Layer | Learned ad-hoc parameters | Orthonormality + KL Divergence |
| *KMEx* | Distance based | Nearest Neighbor | $k$-means | Clustering |

the need for retraining, to improve the prototype positioning on the SEM's embeddings and to better cover the attributes and their correlations.

## 2 Prototypical self-explainable models

In this section, we review the recent literature on PSEMs for the task of image classification, which is the focus of this work, emphasizing their design considerations as well as evaluation approaches.

PSEMs for image classification typically consist of four common components: an encoder, a set of prototypes, a similarity function, and a transparent classifier. The encoder is typically sourced from a black-box model, thereby making the latter the *closest* (to the SEM architecture) natural baseline for comparison until now. Prototypes are class-concepts that live in the embedding space and serve as global class explanations, i.e., representative vectors, that eliminate the necessity to examine the entire dataset for explaining the learning of the model. The similarity function compares features extracted from the input to those embodied in the prototypes. Ultimately, a transparent classifier transforms the similarity scores into class predictions. The fact that the final classification revolves around the prototypes makes them a critical component of SEMs. In addition to these, other modules have also been utilized in the literature to facilitate the learning of prototypes, such as a decoder to align the embedding space to the input space (Parekh et al., 2021; Gautam et al., 2022), or a companion encoder to learn the prototypical space (Parekh et al., 2021).

### 2.1 Predicates for SEMs

PSEMs are designed to learn inherently interpretable global class concepts. Three principles arise from the literature to form a framework for their construction: transparency, diversity, and trustworthiness (Gautam et al., 2022).

- A model is said to be *transparent* if the downstream task involves solely human-interpretable concepts and operations.
- The learned concepts are *diverse* if they capture non-overlapping information in the embedding space and, therefore, in the input space.
- *Trustworthiness* comes in several dimensions. An SEM is deemed *faithful* if its classification accuracy and explanations match its black-box counterpart. In addition, local and global explanations should be *robust* (similar inputs yield similar explanations) and truly reflect the important features of the input with respect to the downstream task.

### 2.2 Related work

Prototypical networks, first introduced by Snell et al. (2017), aim to learn an embedding space structured such that points representing data from the same class are closer together, while points from different classes are further apart. In these networks, 'prototypes' acts as representative centroids for each class within the embedding space. This approach has proven to be effective in scenarios where classifiers need to generalize to new classes that were not encountered during the training phase, a challenge typical of few-shot learning (Snell et al., 2017; Trosten et al., 2023). While the structure learned in PSEMs resembles the class-wise structuring of the embedding space used for few-shot learning, semantic meaning from the input-space is attributed to these prototypes, enhancing their inherent interpretability (Alvarez Melis & Jaakkola, 2018; Chen et al., 2019b). KMEx builds upon this approach and transforms any black-box model into a self-explainable one without the need for restructuring the already learned embedding space.

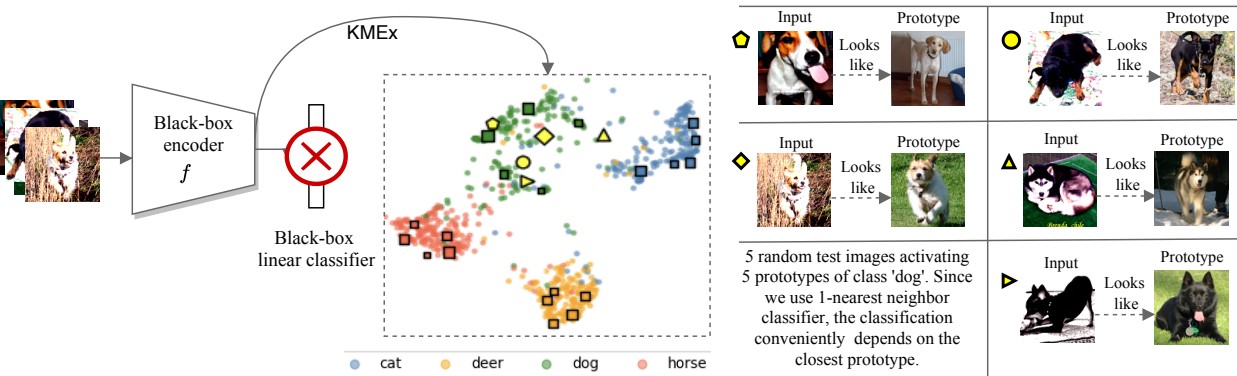

Figure 1: Schematic representation of KMEx. Left: The black-box classifier is removed and replaced by a nearest neighbor classifier based on prototypes learned using $k$-means in the embedding space. The UMAP (McInnes et al., 2018) representation is the projection of the learned embedding space for STL-10, along with prototypes, depicted as squares. Right: The prototypes are visualized in the input space using the closest training images.

The first general framework to compute interpretable concepts was SENN (Alvarez Melis & Jaakkola, 2018), which relies on a complex architecture and loss function to ensure interpretability. Following this, several SEMs have emerged, one of the most popular being ProtoPNet (Chen et al., 2019b). The latter introduces a learnable prototype similarity layer with a fixed number of prototypes per class. Several methods have followed to address the limitations of ProtoPNet. For example, ProtoPShare (Rymarczyk et al., 2021), ProtoTree (Nauta et al., 2021) and ProtoPool (Rymarczyk et al., 2022) proposed learning of shareable prototypes across classes, (Donnelly et al., 2021) proposed adaptive prototypes which change their spatial location based on the input image and TesNet (Wang et al., 2021) introduced a plug-in embedding space spanned by basis concepts constructed on the Grassman manifold, thereby inducing diversity among prototypes.

In parallel to ProtoPNet and its extensions, several other SEMs have been proposed. FLINT (Parekh et al., 2021) introduces an interpreter network with a learnable attribute dictionary in addition to the predictor. SITE (Wang & Wang, 2021) introduces regularizers for obtaining a transformation-equivariant SEM. ProtoVAE (Gautam et al., 2022) learns a transparent prototypical space thanks to a backbone based on a variational autoencoder, thereby having the capability to reconstruct prototypical explanations using the decoder.

While all the existing SEMs have demonstrated effective generation of explanations alongside comparable accuracies, they invariably demand significant architectural modifications and integration of multiple loss functions. This often introduces several additional hyperparameters to achieve satisfactory performance. For instance, in the case of ProtoPNet, a three-step training process involves encoder training, prototype projection for explainability, and last-layer training. Furthermore, as highlighted in FLINT (Parekh et al., 2021), a simultaneous introduction of all losses can lead to suboptimal optimization. Their workaround strategy involves distinct loss combinations for fixed epochs. These intricate training strategies, combined with the challenge of training large deep learning architectures, complicate the accessibility of SEMs, thereby emphasizing the demand for more resource-efficient alternatives. KMEx, a universally applicable method that necessitates no re-training, no additional loss terms for training the backbone, and minimal architectural adjustments for learning the prototypes, presents an efficient solution to this challenge. Considering our general contributions to SEMs, we use ProtoPNet, a representative approach encompassing all its extensions, as a baseline in this work. Additionally, we also consider FLINT and ProtoVAE, which cover the diversity of the SEM's literature in terms of backbones, similarity, and loss functions. A summary of these baselines is given in Table 1, along with KMEx, which is presented in the following section.

# 3 KMEx: a universal explainer

In this section, we introduce KMEx which transforms a black-box model into an SEM, fulfilling all the aforementioned predicates. Note that to enhance legibility, KMEx may refer to both the method and the transformed model in the following.

## 3.1 Notations

We will now present some notations that will be utilized in the subsequent sections. We consider an image dataset $\mathcal{X} = \{(\boldsymbol{x}_i, \boldsymbol{y}_i)\}_{i=1}^N$ made of $N > 0$ images split into $K > 0$ classes, where $\boldsymbol{x}_i \in \mathbb{R}^{W \times H \times C}$ is an image of width $W > 0$, height $H > 0$, and with $C > 0$ channels, and $\boldsymbol{y}_i \in [1 \ldots K]$ encodes its label. Any model in the following contains an encoder $f$ which projects an input $\boldsymbol{x}_i$ into a point $f(\boldsymbol{x}_i) = \boldsymbol{z}_i \in \mathbb{R}^D$ of a $D > 0$ dimension vector space, also referred to as the *embedding space*. We consider a set of $L > 0$ prototypes per class $\{p_{11} \ldots p_{K1} \ldots p_{1L} \ldots p_{KL}\} \subset \mathbb{R}^D$ that are vectors of the embedding space. Finally, we denote as $\mathfrak{s}$ a generic similarity measure between vectors of the embedding space that returns larger values to pairs of vectors deemed similar. This could be the cosine function or the negative of the $\ell_2$ distance.

## 3.2 KMEx

Let us consider a trained model made of an encoder and a classifier. It can be converted into a self-explainable model using the following procedure:

1. Learn $L$ prototypes for each of the $K$ classes using $k$-means on the embedding of the training data of the class.
2. Replace the classifier with a 1-nearest neighbor classifier comparing the embedding of an input with the prototypes using the negative of the $\ell_2$ distance as the similarity measure.

The resulting model is referred to as the K-Means Explainer (KMEx) of the original model. A schematic representation of the operations is depicted in Figure 1. We chose a constant number of prototypes per class in order to ease the notations. However, in practice, more complex classes may require more prototypes than others, which is straightforward to adapt in KMEx. Note that the KMEx conversion is not a post-hoc explainability method per se as what it produces is another model that is inherently interpretable owing to the central role of the prototypes and the transparent classifier. Although the trained encoder is re-used, the KMEx's predictions are computed differently using prototypes that were not part of the initial model.

We further highlight that $k$-means is computed per class and on the embedding space, which usually has a reasonable number of dimensions (512 for ResNet34). Hence, the computational cost is limited and manageable by classic implementations, irrespective of the complexity of the data. For very large datasets, it can be approximated by computing $k$-means on a subset of the training set or by using other efficient implementations (Johnson et al., 2019).

## 3.3 KMEx is an SEM

**Visualisation of explanations**  The explanations for a PSEM are two-fold. *Global* explanations involve visualizations of prototypes in the input space, providing insights into the model's acquired knowledge. *Local* explanations, on the other hand, entail pixel-level explanations for input images, revealing which portions of an image are activated by each prototype. For KMEx, we provide global explanations by visualizing the training images that are closest to the corresponding prototypes in the embedding space. This approximation is justified by the problem solved by $k$-means, which makes it unlikely for a prototype to be out of distribution. For local explanations, we adhere to previous works and employ Prototypical Relevance Propagation (PRP), a technique demonstrated to be efficient and accurate for ProtoPNet (Gautam et al., 2023).
**Transparency**  The nearest prototype classifier of KMEx allows backtracking of the influence of a prototype on the predictions, which relates to a distance in the embedding space, thus embodying transparency.
**Faithfulness of the predictions**  If the original trained model learned to separate well the classes in the embedding, there should be enough inter-class distance for the linear partition of $k$-means to yield KMEx prototypes that also correctly separate the classes and thus achieve classification performance akin to that of

Table 2: Evaluation strategies for the predicates used by state-of-the-art SEMs. The proposed evaluation framework is *italicised*.

| | Transparency | Trustworthiness | | | Diversity |
|---|---|---|---|---|---|
| | | **Baseline** | **Faithfulness** | **Robustness** | |
| ProtoPNet | Visualization | Black-box | Accuracy | - | - |
| FLINT | Visualization | Black-box | Accuracy | - | - |
| ProtoVAE | Visualization | Black-box/SEM | Accuracy | AI/AD/RO | Reconstruction visualization |
| *Proposed Evaluation* | *Ghosting* | Black-box/SEM/ *KMEx* | *Accuracy/ KL Divergence* | Area under RO curve | *Inter-prototype similarity* |

the trained model.

**Faithfulness and robustness of the explanations**   The only difference between a black-box and its KMEx is how the predictions are derived from the embedding. Therefore, considering identical weights in both models' encoders, most of the operations involved in the generation of local explanation maps are common to both, thus similar explanations with similar level of robustness are expected, regardless of the technique chosen to generate local explanations.

**Diversity**   The purpose of the prototypes is to serve as representatives of their neighborhood in the embedding space. The diversity predicate implicitly requires that they also spread over the embedding. To satisfy this predicate without compromising their function, we aim to position the prototypes on the accumulation points of the embedding. These are captured as the modes of a Gaussian density estimate. Computing such a model for a high dimensional and sparse dataset is costly, hence we approximate it using $k$-means. Finally, given that $k$-means employs a uniform prior on the cluster probabilities, this method has the advantage of covering as much of the data in the embedding space as possible, thus fostering diversity.

## 4   Evaluations

As stated earlier, existing SEMs build upon three shared predicates but adopt varied strategies to ensure their fulfillment. *Transparency* is assumed based on architectural choices and, at best, confirmed through visualization of prototypes using different strategies, such as upsampling (Chen et al., 2019b), activation maximization (Parekh et al., 2021; Mahendran & Vedaldi, 2016) and PRP (Gautam et al., 2023), accompanied with similarity scores. The *trustworthiness* predicate is the most quantifiable one. The faithfulness of the performance with respect to the "closest" black-box is often reduced to a comparison of accuracies, and the robustness of the explanations is evaluated via recent measures such as Average Increase (AI), Average Drop (AD), and the area under the Relevance Ordering curve (AUROC) test (Lee et al., 2021; MacDonald et al., 2019; Hedström et al., 2022). Nonetheless, the quantification of disparities between local explanations generated by an SEM and its nearest black-box model has been largely disregarded. We emphasize that this aspect grows in significance, particularly as we transition to techniques that transform existing black-box models into interpretable ones without re-training, a domain where KMEx stands as the first approach. Finally, prototypical *diversity* has been largely overlooked in prior research, with evaluations, if conducted, being primarily qualitative in nature (Gautam et al., 2022).

In this section, we first evaluate KMEx following the evaluation protocols used in the original papers of the selected baselines, which are summarized in Table 2. Following this, we propose our full quantitative evaluation framework based on the predicates for SEMs, highlighting the gaps in the evaluation of SEMs existing until now, also summarized in Table 2. Additionally, we present a quantitative study of the diversity and subclass representation captured by the prototypes learned by existing SEMs and their KMEx counterparts.

### 4.1   Datasets, implementation and baselines

We evaluate all methods on 7 datasets, MNIST (Lecun et al., 1998), FashionMNIST (Xiao et al., 2017) (fMNIST), SVHN (Netzer et al., 2011), CIFAR-10 (Krizhevsky, 2009), STL-10 (Coates et al., 2011), a subset

Table 3: Prediction accuracy for SEMs demonstrating the effectiveness of KMEx as an SEM baseline. Reported numbers are averages over 5 runs along with standard deviations. Statistically larger values are highlighted in **bold**.

| | MNIST | fMNIST | SVHN | CIFAR-10 | STL-10 | QuickDraw | CelebA |
|---|---|---|---|---|---|---|---|
| ResNet34 | $\mathbf{99.4^{\pm 0.0}}$ | $\mathbf{92.4^{\pm 0.1}}$ | $92.6^{\pm 0.2}$ | $\mathbf{85.6^{\pm 0.1}}$ | $\mathbf{91.8^{\pm 0.1}}$ | $86.5^{\pm 0.1}$ | $98.5^{\pm 0.0}$ |
| FLINT | $99.2^{\pm 0.1}$ | $91.8^{\pm 0.5}$ | $91.1^{\pm 0.7}$ | $82.2^{\pm 1.1}$ | $87.5^{\pm 0.6}$ | $87.3^{\pm 0.2}$ | $97.2^{\pm 0.3}$ |
| ProtoPNet | $\mathbf{99.4^{\pm 0.1}}$ | $\mathbf{92.4^{\pm 0.2}}$ | $\mathbf{94.4^{\pm 0.1}}$ | $84.9^{\pm 0.2}$ | $88.1^{\pm 0.6}$ | $\mathbf{87.8^{\pm 0.2}}$ | $98.1^{\pm 0.0}$ |
| ProtoVAE | $\mathbf{99.4^{\pm 0.0}}$ | $92.7^{\pm 0.5}$ | $93.8^{\pm 0.6}$ | $83.0^{\pm 0.2}$ | $85.6^{\pm 1.1}$ | $85.1^{\pm 0.8}$ | $\mathbf{98.6^{\pm 0.0}}$ |
| *R34+KMEx* | $\mathbf{99.4^{\pm 0.0}}$ | $\mathbf{92.3^{\pm 0.1}}$ | $92.4^{\pm 0.1}$ | $85.3^{\pm 0.1}$ | $\mathbf{91.9^{\pm 0.2}}$ | $86.6^{\pm 0.2}$ | $98.3^{\pm 0.0}$ |

of QuickDraw (Parekh et al., 2021) and binary classification for male and female for the CelebA dataset (Liu et al., 2015). We use a vanilla ResNet34 (He et al., 2016) as the encoder for all the models and fix the number of prototypes per class as 20 for CelebA and 5 for all other datasets. Further implementation details are provided in Appendix A.2. For baselines, we train ProtoPNet (Chen et al., 2019b), FLINT (Parekh et al., 2021), and ProtoVAE (Gautam et al., 2022) for learning image-level prototypes. For ProtoPNet, we use average pooling to generate image-level prototypes. For FLINT, we use the interpreter network FLINT-$g$.

## 4.2  Traditional evaluation of KMEx

In this section, we evaluate KMEx following previous lines of works (Wang & Wang, 2021; Gautam et al., 2022). We start with comparing the predictive performance of KMEx, which is then followed by an evaluation of explanations consisting of visualization of prototypes and evaluating the robustness of explanations. In Appendix A.7, we present preliminary results for a patch-based KMEx.

### 4.2.1  Predictive performance

We report the accuracy achieved by KMEx, as well as selected baselines in Table 3. As can be observed, with a maximal discrepancy of 0.3 points, KMEx performs on par with its corresponding ResNet34 black-box base model, thereby validating the change of classifier. On the other hand, FLINT performs consistently and significantly worse than the black-box on all the datasets. ProtoPNet and ProtoVAE outperform ResNet34 and R34+KMEx on two of the datasets.

### 4.2.2  Evaluation of explanations

In previous works, the evaluation of explanations was twofold: 1) qualitative evaluation of prototypes and 2) evaluation of robustness of prototypical explanations.

**Qualitative evaluation**  We visualize prototypes learned by KMEx for MNIST and STL-10 datasets in Figure 2 (top row). Additional visualizations for other datasets are given in Appendix A.8.1 and A.8.2. We demonstrate the "*this* looks like *that*" behavior exhibited by KMEx for test images in the bottom row of Figure 2, along with their corresponding PRP maps, demonstrating the regions activated in the test images by their closest prototypes. As observed, for the MNIST dataset, the activations are in response to the shape of the digit in the prototype. Similarly, for STL-10, the closest prototype has emphasized key features of a bird, such as the head, beak, and eyes, as well as a portion of the sky in the background.

**Robustness evaluation**  We evaluate the robustness of the local explanations using the area under the relevance order curve (AUROC) Schulz et al. (2020); MacDonald et al. (2019). The relevance order (RO) curve is the evolution of the probability of the predicted class as the least relevant pixels are progressively replaced by white noise. The relevance derives from PRP local explanations computed with respect to the most similar prototypes. For the black-box model, we rely instead on the Layer-wise Relevance Propagation (Bach et al., 2015) (LRP) explanations with respect to the class probability. The local explanations are computed using the Zennit package Anders et al. (2021) with `EpsilonPlusFlat` composer.

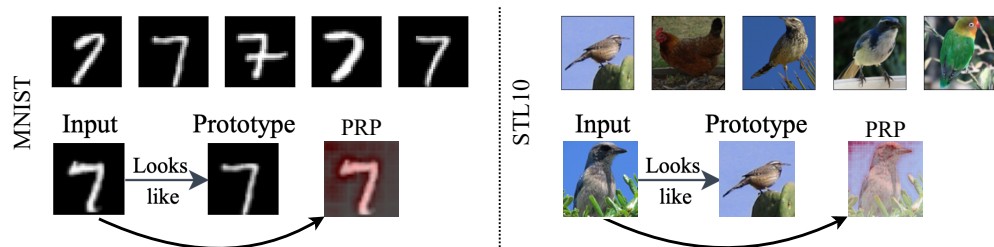

Figure 2: Qualitative evaluation of KMEx: Prototypes learned by KMEx for MNIST for class '7' (left) and STL-10 for class 'bird' (right) are shown at the top, demonstrating global explainability. *This* looks like *that* behavior for test images are shown at the bottom, along with PRP maps demonstrating the regions activated by closest prototypes (in red) for the test images, exhibiting local explainability.

Table 4: Area under the Relevance Order curves evaluate the robustness of the PRP local explanations each model produces. Reported numbers are averages over 100 test images along with standard deviations. Statistically larger values are highlighted in **bold**.

| | | | | | | | |
|---|---|---|---|---|---|---|---|
| ResNet34 | $0.397^{\pm 0.000}$ | $0.360^{\pm 0.000}$ | $0.318^{\pm 0.000}$ | $\mathbf{0.354^{\pm 0.000}}$ | $\mathbf{0.466^{\pm 0.000}}$ | $\mathbf{0.688^{\pm 0.000}}$ | $0.671^{\pm 0.000}$ |
| FLINT | $0.116^{\pm 0.061}$ | $0.142^{\pm 0.082}$ | $0.098^{\pm 0.026}$ | $0.133^{\pm 0.065}$ | $0.090^{\pm 0.055}$ | $0.108^{\pm 0.048}$ | $0.484^{\pm 0.073}$ |
| ProtoPNet | $\mathbf{0.572^{\pm 0.040}}$ | $0.339^{\pm 0.012}$ | $0.388^{\pm 0.032}$ | $0.313^{\pm 0.013}$ | $\mathbf{0.445^{\pm 0.046}}$ | $0.667^{\pm 0.005}$ | $\mathbf{0.698^{\pm 0.107}}$ |
| ProtoVAE | $0.200^{\pm 0.020}$ | $0.291^{\pm 0.040}$ | $\mathbf{0.680^{\pm 0.020}}$ | $0.332^{\pm 0.012}$ | $0.113^{\pm 0.001}$ | $0.129^{\pm 0.001}$ | $\mathbf{0.832^{\pm 0.060}}$ |
| *R34+KMEx* | $0.401^{\pm 0.000}$ | $\mathbf{0.379^{\pm 0.000}}$ | $0.317^{\pm 0.000}$ | $0.347^{\pm 0.000}$ | $0.454^{\pm 0.000}$ | $0.686^{\pm 0.000}$ | $0.616^{\pm 0.000}$ |

Curves averaged over 100 test images and all prototypes for each model on MNIST, CIFAR-10 and CelebA are depicted in Figure 3. For reference, the plots also show the respective average RO curves when the pixels are masked at random. The datasets were chosen for the different behaviors they induce in the tested models. In the three cases, FLINT's (blue) curves are relatively flat. On MNIST, the three other models show the three characteristic possible behaviors. ProtoVAE (green) shows poor robustness as the curve decreases sharply early on and remains below its random curve (dashed green). ResNet34 (gray) and its R34+KMEx (red) decrease somehow linearly before flattening as the proportion of masked pixels reaches 0.7. On the other hand, the explanations of ProtoPNet (yellow) demonstrate some robustness on MNIST as its RO curve remains high for longer before starting to decrease. On CIFAR-10, all the models but FLINT present a similar, not-so-robust behavior. Finally, ProtoVAE provides the most robust explanations as its curve remains above all the other curves.

These observations are reflected in Table 4, where we report averages AUROC with and standard deviations for 100 test images on each dataset. On MNIST, ProtoPNet returns the largest value. On CIFAR-10, all but FLINT report similar values despite being statistically different. On CelebA, ProtoVAE reports the largest AUROC, followed by ProtoPNet. Overall, R34+KMEx returns AUROC close to ResNet34, its black-box base model. Yet the differences are always significant given the very low standard deviations. These results suggest that KMEx does not produce more robust explanations on its own. This is anticipated as KMEx aims to facilitate the interpretation of a learned latent representation but does not enforce robustness or stability of the prototypes during the training process, unlike other SEMs.

### 4.3 Quantitative evaluation of SEMs

Having demonstrated the traditional evaluation of the proposed KMEx, we now address the lack of a comprehensive evaluation framework for SEMs that quantitatively evaluates the predicates. First, we expose for the first time how transparency is often undermined by unused prototypes (ghosting) and measure the phenomenon. Next, we objectively quantify the faithfulness of local explanations and the diversity of the prototypes without resorting to visual inspection. Again, the rationale is to propose a framework to assess objectively each model's strengths and weaknesses.

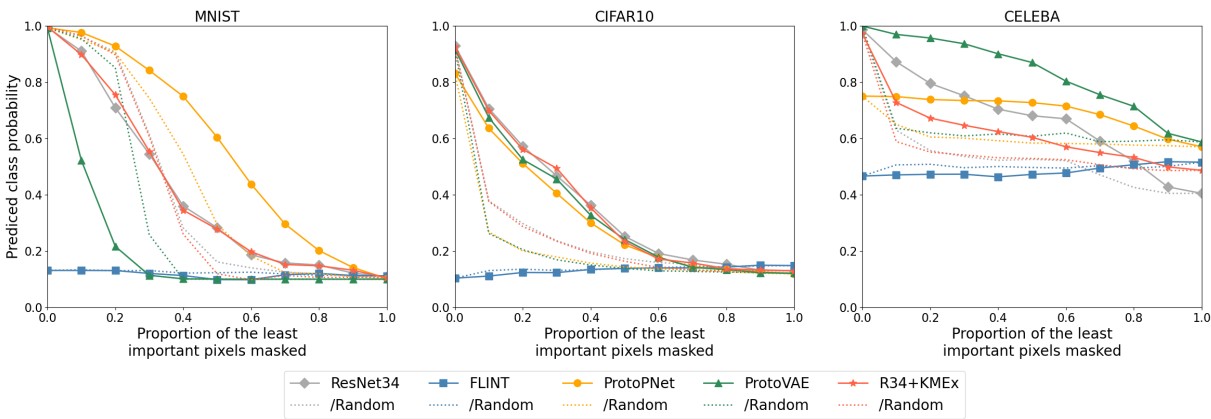

Figure 3: Relevance Ordering curves computed on different datasets and with different architectures, along with the respective random baselines (dashed).

### 4.3.1 Transparency and concept ghosting

The transparency predicate allows the user to backtrack the influence of the learned concepts on the predictions and is usually enforced through architecture design. However, we observe for state-of-the-art SEMs that, in practice, some learned prototypes are not reachable from the predictions. More specifically, they are never *activated* by any training data point of their class, i.e., they are never the most similar prototype of any training data. This so-called *ghosting* of the prototypes not only gives a false sense of needed concepts but also undermines the notion of transparency itself, as the link between prototypes and prediction can not be fully trusted.

In the case of a distance-based similarity measure (Gautam et al., 2022; Chen et al., 2019b), if $k = \mathrm{argmax}_l \, \mathfrak{s}(z_i, p_l)$, then prototype $p_k$ is the closest to $z_i$. If several points activate $p_k$, this indicates that the data embedding aggregates around a concentration point close to $p_k$. In contrast, if $p_k$ never maximizes any $\mathfrak{s}(z_i, \cdot)$, then there might be no data in its neighborhood. The prototype is either out-of-distribution or lies in an area of low density. In the case of a dot-product-based similarity measure (Parekh et al., 2021), if $p_k$ is activated by $z_i$, then $z_i$ and $p_k$ are aligned. Assuming that $z_i$ is not the only maximizer of $\mathfrak{s}(\cdot, p_k)$, then $p_k$ carries a direction along which the data accumulates. In contrast, if $p_k$ never maximizes any $\mathfrak{s}(z_i, \cdot)$, then the data does spread along its direction and may represent either a variation within a class or, in the worst case, noise.

We propose to quantify this *ghosting* phenomenon based on the average activation frequencies over the prototypes on the training set:

$$\mathcal{D}_{\mathrm{tsp}} = 1 - \frac{1}{KL} \sum_{k=1}^{K} \sum_{l=1}^{L} [[\exists i \in [1 \ldots N] : \underset{\substack{1 \le u \le K \\ 1 \le v \le L}}{\mathrm{argmax}} \big( \mathfrak{s}(z_i, p_{uv}) \big) = (k, l)]], \tag{1}$$

where $[[\cdot]]$ is the Iverson bracket which returns 1 if the contained statement is true and 0 otherwise. The values of $\mathcal{D}_{\mathrm{tsp}}$ range between 0 and 1, with lower values indicating less ghosting.

In Table 5, we report average $\mathcal{D}_{\mathrm{tsp}}$ scores (Equation 1) with standard deviation over five runs. We observe that ghosting affects all models but, unsurprisingly, never KMEx. Indeed, for such a low number of prototypes, relative to the size of the data, $k$-means is unlikely to create an empty cluster. Interestingly, ProtoVAE also does not ghost any prototype on three out of the seven datasets, suggesting that SEMs with geometrical constraints are more robust to ghosting.

### 4.3.2 Trustworthiness and faithfulness

According to its definition, the trustworthiness predicate encompasses two major axes. The first is the *faithfulness of the predictions*, which we have quantified in terms of accuracies in Table 3. The second aspect

Table 5: *Transparency*: frequency of ghosted prototypes by SEMs.Statistically smaller values are highlighted in **bold**.

| | MNIST | fMNIST | SVHN | CIFAR-10 | STL-10 | QuickDraw | CelebA |
|---|---|---|---|---|---|---|---|
| FLINT | $0.246^{\pm 0.148}$ | $\mathbf{0.251^{\pm 0.258}}$ | $0.060^{\pm 0.025}$ | $0.174^{\pm 0.066}$ | $0.156^{\pm 0.070}$ | $\mathbf{0.230^{\pm 0.360}}$ | $\mathbf{0.220^{\pm 0.365}}$ |
| ProtoPNet | $0.610^{\pm 0.028}$ | $0.580^{\pm 0.061}$ | $0.490^{\pm 0.055}$ | $0.165^{\pm 0.051}$ | $0.221^{\pm 0.081}$ | $0.156^{\pm 0.046}$ | $0.695^{\pm 0.099}$ |
| ProtoVAE | $\mathbf{0.000^{\pm 0.000}}$ | $\mathbf{0.012^{\pm 0.016}}$ | $0.012^{\pm 0.010}$ | $\mathbf{0.000^{\pm 0.000}}$ | $0.642^{\pm 0.043}$ | $0.561^{\pm 0.074}$ | $0.335^{\pm 0.068}$ |
| *R34+KMEx* | $\mathbf{0.000^{\pm 0.000}}$ | $\mathbf{0.000^{\pm 0.000}}$ | $\mathbf{0.000^{\pm 0.000}}$ | $\mathbf{0.000^{\pm 0.000}}$ | $\mathbf{0.000^{\pm 0.000}}$ | $\mathbf{0.000^{\pm 0.000}}$ | $\mathbf{0.000^{\pm 0.000}}$ |

Table 6: *Faithfulness of explanations*: divergence of LRP explanation maps from the black-box. Reported numbers are average over 100 test images. Statistically smaller values are highlighted in **bold**.

| | MNIST | fMNIST | SVHN | CIFAR-10 | STL-10 | QuickDraw | CelebA |
|---|---|---|---|---|---|---|---|
| FLINT | $0.696^{\pm 0.041}$ | $0.883^{\pm 0.035}$ | $0.790^{\pm 0.034}$ | $0.739^{\pm 0.054}$ | $2.468^{\pm 0.115}$ | $0.783^{\pm 0.050}$ | $4.411^{\pm 0.198}$ |
| ProtoPNet | $0.749^{\pm 0.045}$ | $1.071^{\pm 0.084}$ | $0.906^{\pm 0.034}$ | $0.896^{\pm 0.046}$ | $2.598^{\pm 0.109}$ | $0.898^{\pm 0.048}$ | $4.785^{\pm 0.106}$ |
| ProtoVAE | $0.656^{\pm 0.005}$ | $0.667^{\pm 0.019}$ | $0.431^{\pm 0.015}$ | $0.605^{\pm 0.019}$ | $2.441^{\pm 0.111}$ | $0.716^{\pm 0.100}$ | $4.268^{\pm 0.301}$ |
| *R34+KMEx* | $\mathbf{0.130^{\pm 0.000}}$ | $\mathbf{0.105^{\pm 0.000}}$ | $\mathbf{0.115^{\pm 0.000}}$ | $\mathbf{0.117^{\pm 0.000}}$ | $\mathbf{0.461^{\pm 0.000}}$ | $\mathbf{0.088^{\pm 0.000}}$ | $\mathbf{0.717^{\pm 0.000}}$ |

concerns the *robustness of the explanations*, which we measured using RO curves (MacDonald et al., 2019; Schulz et al., 2020) in Table 4. An often overlooked aspect of the trustworthiness predicate is the *faithfulness of the explanations*. Indeed, SEMs differ from black-box models in their architecture and training and, therefore, also in their local explanations. However, as we move towards methods that convert black-box models into self-explainable, it becomes crucial to quantitatively evaluate this discrepancy. We propose to use the Kullback-Leibler (KL) divergence ($D_{KL}$) between the Layer-wise Relevance Propagation (LRP) maps (Bach et al., 2015) for the prediction probabilities produced by the SEM and the black-box baseline. Since the divergence acts on distributions, the relevance maps need to be normalized. The use of LRP aligns with the previous utilization of PRP. Other methods could be used, yet our intention here is not to evaluate the SEMs with respect to these methods but rather with respect to the predicates.

Let us denote the output of the local explanation method for an input $x$ as $\mathfrak{e}(x) \in \mathbb{R}^{W \times H \times C}$. The corresponding normalized relevance $\mathfrak{e}_n(x) \in \mathbb{R}^{W \times H}$ is defined as:

$$\mathfrak{e}_n(x) = \frac{\max_{c=1\ldots C} |\mathfrak{e}(x)|(\cdot, \cdot, c)}{\sum_{w=1\ldots W} \sum_{h=1\ldots H} \max_{c=1\ldots C} |\mathfrak{e}(x)|(w, h, c)} \tag{2}$$

The divergence of $\mathfrak{e}_n(x)$ with respect to the normalized local explanation maps produced by the black-box backbone $\mathfrak{e}_n^{\text{bbox}}(x)$ is measured by $\mathcal{D}_{\text{fdl}}$ defined as follows:

$$\mathcal{D}_{\text{fdl}} = \sum_{w=1}^{W} \sum_{h=1}^{H} \mathfrak{e}_n(x)(w, h) \log \left( \frac{\mathfrak{e}_n(x)(w, h)}{\mathfrak{e}_n^{\text{bbox}}(x)(w, h)} \right) \tag{3}$$

The KL divergence is zero if, and only if, the distributions are equal. Consequently, $\mathcal{D}_{\text{fdl}}$ can be null if, and only if, the SEM and the black-box models always produce the same explanations.

In Table 6, we report the average $\mathcal{D}_{\text{fdl}}$ based on LRP and standard deviation for each SEM on 100 images of each dataset. Normalized LRP maps for each model and on different datasets are represented in Appendix A.4. Again, KMEx produces the most faithful feature importance maps, which is expected since most of the operations happen in the encoder, which originates from the black-box model.

### 4.3.3 Interpreted diversity

The abundance of existing strategies to guarantee the *diversity* of an SEM reflects the subjectivity of the notion. Thus, it is not obvious how to evaluate this predicate in the input space, especially given that very few public image datasets provide attributes describing the image. Thus, the evaluation has to be done in the embedding space. However, using a metric based on distances in the embedding space would disadvantage

Table 7: *Interpreted diversity*: quantitative evaluation of diversity for different SEMs. Statistically larger values are highlighted in **bold**.

|  | MNIST | fMNIST | SVHN | CIFAR-10 | STL-10 | QuickDraw | CelebA |
|---|---|---|---|---|---|---|---|
| FLINT | $0.991^{\pm 0.003}$ | $0.991^{\pm 0.003}$ | $0.967^{\pm 0.017}$ | $0.963^{\pm 0.018}$ | $0.999^{\pm 0.000}$ | $0.968^{\pm 0.010}$ | $0.948^{\pm 0.013}$ |
| ProtoPNet | $0.172^{\pm 0.017}$ | $0.160^{\pm 0.021}$ | $0.173^{\pm 0.018}$ | $0.233^{\pm 0.014}$ | $0.120^{\pm 0.035}$ | $0.241^{\pm 0.029}$ | $0.715^{\pm 0.021}$ |
| ProtoVAE | $0.014^{\pm 0.000}$ | $0.014^{\pm 0.000}$ | $0.014^{\pm 0.000}$ | $0.014^{\pm 0.000}$ | $0.013^{\pm 0.000}$ | $0.014^{\pm 0.000}$ | $\mathbf{0.014^{\pm 0.000}}$ |
| *R34+KMEx* | $\mathbf{0.000^{\pm 0.000}}$ | $\mathbf{0.000^{\pm 0.000}}$ | $\mathbf{0.000^{\pm 0.000}}$ | $\mathbf{0.000^{\pm 0.000}}$ | $\mathbf{0.000^{\pm 0.000}}$ | $\mathbf{0.000^{\pm 0.000}}$ | $0.026^{\pm 0.001}$ |

methods relying on a dot-product-based similarity measure and the other way around. We therefore propose to evaluate SEMs on the basis of their own interpretation of diversity and to base our *diversity* metric on the models' own similarity function. In other words, the idea is to assess the extent to which models achieve diversity on the basis of their own model choices.

The overarching objective of existing approaches for diversity is to prevent prototype collapse. In such a case, the information captured by the prototypes highly overlaps, yielding inter-prototype similarities ($\mathfrak{s}(p_{kl}, p_{uv})$ with $(k,l) \neq (u,v)$) as high as prototypical self-similarities ($\mathfrak{s}(p_{kl}, p_{kl})$). On the other hand, if prototype collapse is well alleviated, the inter-prototype similarities are low, while the self-similarities remain high. This observation motivates the use of the entropy function.

Accordingly, we quantify the diversity of a set of concepts using $\mathcal{D}_{\mathrm{dvs}}$ defined as the average of the normalized entropy of the similarities between each prototype. The computation compares each prototype to all the others without discarding the ghosted prototypes, as they may indicate a collapse.

$$\mathcal{D}_{\mathrm{dvs}} = \frac{1}{KL\log(KL)} \sum_{k=1}^{K} \sum_{l=1}^{L} \mathbf{H}\left(\mathrm{Softmax}\left(\langle \mathfrak{s}(p_{kl}, p_{uv}) \rangle_{\substack{1 \leq u \leq K \\ 1 \leq v \leq L}}\right)\right), \quad (4)$$

where $\mathbf{H}$ is the entropy function. The $\log(KL)$-normalization restricts the measure to $[0,1]$ and allows comparisons between different runs, number of prototypes, as well as models. Large values indicate more similarity between the clusters and, thereby, less diversity.

In Table 7, we report the average $\mathcal{D}_{\mathrm{dvs}}$ (Equation 4) score with standard deviations over five runs for each SEM on each dataset. Recall that $\mathcal{D}_{\mathrm{dvs}}$ estimates how well a model satisfies its own interpretation of diversity. Following this, KMEx and FLINT are, respectively, the most and the least satisfying models. We emphasize here that a low diversity doesn't reflect the caliber of the learned embedding space and only suggests an important overlap of information between the representative prototypes learned for the embeddings.

### 4.3.4  Summary

We summarize the quantitative assessment presented as a radar plot in Figure 4. The axes correspond, respectively, to the average results over the datasets presented in Tables 3, 4, 5, 6 and 7. The axes are set such that a larger polygonal area is better. The transparency (Transp.) axis ranges from 1 in the center to 0 on the outer circle. The faithfulness of accuracy (Faith.Acc.) corresponds to the difference from that of the black-box base model. Along that axis, points on the outer circle mean equal accuracy with the black-box base model. models below perform worse, with the center indicating a difference of 5 points. Models performing better have their points beyond the outer circle. The faithfulness of the explanation (Faith.Expl.) axis is restricted to $\mathcal{D}_{\mathrm{fdl}}$ ranging from 0 on the outer circle to 2 on the center. A better robustness of explanation (Rob.Expl.) means a value closer to 1. The worst possible value is 0. Finally, the diversity score (Div.) ranges from 1 on the outer circle to 0 on the center. Plots for individual datasets are presented in Figure 7 in the appendix.

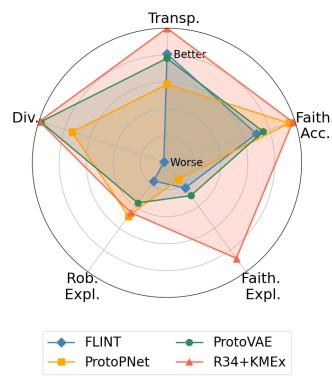

Figure 4: Summary of each model's strengths and weaknesses.

Table 8: Quantitative evaluation of diversity by applying KMEx to learned SEM embeddings. Statistically significance difference with respect to Table 7 are marked in **bold**

|  | MNIST | fMNIST | SVHN | CIFAR-10 | STL-10 | QuickDraw | CelebA |
|---|---|---|---|---|---|---|---|
| FLINT+KMEx | $0.455^{\pm 0.136}$ | $0.456^{\pm 0.232}$ | $0.513^{\pm 0.135}$ | $0.358^{\pm 0.097}$ | $0.530^{\pm 0.118}$ | $0.406^{\pm 0.226}$ | $0.520^{\pm 0.273}$ |
| Difference | $\mathbf{-0.536}$ | $\mathbf{-0.535}$ | $\mathbf{-0.454}$ | $\mathbf{-0.605}$ | $\mathbf{-0.469}$ | $\mathbf{-0.562}$ | $\mathbf{-0.427}$ |
| ProtoPNet+KMEx | $0.335^{\pm 0.008}$ | $0.237^{\pm 0.015}$ | $0.276^{\pm 0.009}$ | $0.257^{\pm 0.017}$ | $0.028^{\pm 0.003}$ | $0.293^{\pm 0.008}$ | $0.717^{\pm 0.019}$ |
| Difference | $\mathbf{0.163}$ | $\mathbf{0.077}$ | $\mathbf{0.104}$ | $0.024$ | $\mathbf{-0.092}$ | $\mathbf{0.052}$ | $0.003$ |
| ProtoVAE+KMEx | $0.014^{\pm 0.000}$ | $0.014^{\pm 0.000}$ | $0.014^{\pm 0.000}$ | $0.014^{\pm 0.000}$ | $0.019^{\pm 0.001}$ | $0.016^{\pm 0.000}$ | $0.015^{\pm 0.000}$ |
| Difference | $0.000$ | $0.000$ | $0.000$ | $0.000$ | $\mathbf{0.005}$ | $\mathbf{0.002}$ | $\mathbf{0.001}$ |

This visualization makes it easy to identify the strengths and weaknesses of each SEM and thus determine the most suitable model according to the problem statement at hand. KMEx suffers the least of ghosting (good transparency) and is the most faithful model with respect to the original black-box both in terms of accuracy and explanation. ProtoPNet performs well in terms of predictions and robustness of the local explanations, but it underperforms in terms of diversity and transparency. This is due to the lack of an inter-class diversity constraint in the ProtoPNet's design. On the other hand, ProtoVAE leads in terms of diversity, but its explanations often differ from the black-box base model's. This is likely due to the utilization of a VAE backbone, which deviates a lot from the architecture of the black-box baseline. Finally, FLINT produces the smallest area. If, on average, it performs on par with ProtoVAE in terms of transparency and faithfulness of the accuracies, it lags behind in all the other measures, especially in terms of diversity.

### 4.4 Diversity and embedding

In this section, we show that for the same embedding learned by an SEM, the KMEx paradigm for prototypes may also be used to improve both the measured and qualitative diversity without retraining the embedding.

**KMEx improves measured diversity** We evaluate first how changing the paradigm of an SEM to KMEx may improve the quantified diversity ($\mathcal{D}_{\mathrm{dvs}}$). We report in Table 8 average scores and standard deviations over five runs for the KMEx of each SEM baseline and the average difference with Table 7. The change of paradigm equips the embedding of FLINT with more quantitatively diverse prototypes. For ProtoPNet, the change is worst on all but one dataset. As for ProtoVAE, which learned prototypes with a very low $\mathcal{D}_{\mathrm{dvs}}$, the changes are very small. Recall that $\mathcal{D}_{\mathrm{dvs}}$ can only serve as an internal evaluation. Therefore, any further analysis of the prototypes requires an external criterion. A visual comparison of the prototypes is presented in Appendix A.8.3.

**KMEx improves minority subclass representation** We further study here the diversity of the prototypes in light of the representation of the attributes they capture. We interpret the notion of a fair subclass representation for SEMs as whether prototypes are able to capture the information about the underrepresented subclasses. For this experiment, we trained ResNet34+KMEx, ProtoPNet, ProtoVAE, FLINT, and their KMEx on the CelebA dataset for male and female classification with varying numbers of prototypes. Prototypes are represented in the input space by their nearest training images, which come with 40 binary attributes as annotations.

To put the observations in the other figures into perspective., we plot first in Figure 5.a the number of prototypes ghosted against the number of prototypes trained. We see here clearly the depth of the issue for FLINT and ProtoPNet. The two following plots (Figure 5.b) depict the number of captured attributes given a number of trained prototypes, including ghosted ones. The left plot shows the results for the baselines and the right one for their KMEx. FLINT starts and ends with fewer captured attributes, and it seems unstable with a large number of prototypes. As for ProtoPNet, it caps at 32 attributes when trained with 12 or more prototypes. On the other hand, the KMEx of any method (right plot), including ResNet34+KMEx (red), always captures more attributes as the number of prototypes increases. The last experiment aims to evaluate how many combinations of attributes are captured using the mean absolute error (MAE) between the attribute

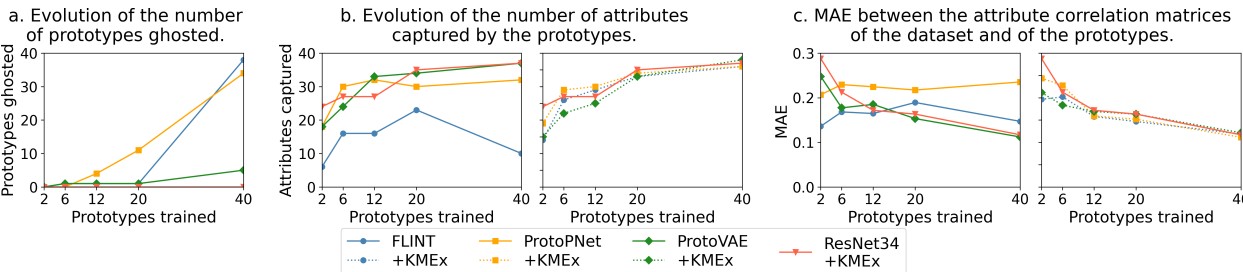

Figure 5: Analysis of the attributes captured by SEMs for different numbers of prototypes for CelebA.

correlation matrices computed from the training set and the prototypes. (Figure 5.c). The correlations based on ProtoPNet's prototypes are the most divergent, whereas ProtoVAE and ResNet34+KMEx consistently come closer to the ground truth as the number of prototypes increases. Again, the attribute correlation computed for any KMEx consistently improves as more prototypes are available.

Overall, KMEx of FLINT improves the most its original model in both criteria: the number of captured attributes and the faithfulness of the attributes correlations. This observation reinforces the intuition that FLINT learns an embedding with much more potential in terms of global explanations than it is able to leverage through the prototypes it learns and its similarity measure.

## 5 Conclusion

In this paper, we introduce KMEx, the first approach for making any black-box model self-explainable. KMEx is a universally applicable, simple, and resource-efficient method that, unlike existing methodologies, does not require re-training of the black-box model. Furthermore, we reconsider the subjective evaluation practices for SEMs by introducing a quantitative evaluation framework that facilitates objective comparisons among SEM approaches. The proposed framework adopts a set of novel metrics to quantify how well SEMs adhere to the established predicates. An extensive evaluation with the help of the proposed framework highlights the strengths and weaknesses of existing SEM approaches when compared to the models obtained from KMEx. This work, therefore, additionally serves as a foundational step towards an objective, comprehensive, and resource-efficient advancement of the SEM field.

One notable limitation of the proposed KMEx is its reliance on selecting a priori the number of prototypes, a characteristic it shares with current state-of-the-art SEMs (Parekh et al., 2021; Gautam et al., 2022; Chen et al., 2019b). Additionally, note that the proposed detailed quantitative evaluation framework is meant to provide an additional perspective and not replace qualitative evaluations of SEMs, which are still required due to the subjective nature of explanations.

## 6 Acknowledgments

This work was financially supported by the Research Council of Norway (RCN), through its Centre for Research-based Innovation funding scheme (Visual Intelligence, grant no. 309439), and Consortium Partners. The work was further partially funded by RCN FRIPRO grant no. 315029 and by the German Ministry for Education and Research (BMBF) through the project Explaining 4.0 (ref. 01IS200551).

We would like to thank Ulf Brefeld for granting us access to the Leuphana University cluster. Further, we would also like to formally acknowledge the reviewers for their invaluable input. Their advice, time, and efforts have significantly aided in enhancing the overall quality of this work.

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

## A  Appendix

In this section, we provide additional details for the proposed evaluation framework. First, we provide additional datasets and implementation details, followed by reporting runtimes for different SEMs used as baselines in this work. We then provide qualitative results for the *faithfulness* of explanations for completeness. This is followed by a detailed summary plot for quantitative evaluation for each dataset. We then provide an ablation study with respect to the similarity measure, number of prototypes per class, and clustering algorithms. Finally, we provide preliminary results on a more complex dataset CUB200 (Welinder et al., 2010), followed by additional qualitative results for KMEx.

### A.1  Dataset details

All datasets used in this work are open-source. For all datasets, we use the official training and testing splits, except for QuickDraw (Ha & Eck, 2018) for which we use a subset of 10 classes that was created by (Parekh et al., 2021). This subset consists of the following 10 classes: *Ant, Apple, Banana, Carrot, Cat, Cow, Dog, Frog, Grapes, Lion.* Each of the classes contains 1000 images of size 28×28 out of which 80% are used for training and the remaining 20% for testing. The MNIST (Lecun et al., 1998), fMNIST (Xiao et al., 2017), CIFAR-10 (Krizhevsky, 2009) datasets consist of 60,000 training images and 10,000 test images of size 28×28, 28×28 and 32×32, respectively. The MNIST, fMNIST and QuickDraw images are resized to 32×32 to obtain a consistent latent feature size. SVHN (Netzer et al., 2011) consists of 73,257 training images and 26,032 images for testing of size 32×32. STL-10 (Coates et al., 2011) consists of 5000 images for training and 8000 for testing of size 96×96. All datasets have 10 classes, except for CelebA (Liu et al., 2015) for which we perform binary classification of male vs female. The number of training and testing images for CelebA are 162,770 and 19,962, respectively, of size 224×224. The licenses for the datasets are provided in Table 9.

For preprocessing, every dataset's respective mean and standard deviation for training data is used for normalization. For MNIST, fMNIST, SVHN and QuickDraw, no augmentations were performed. For STL-10, CIFAR-10, and CelebA, we applied a horizontal flip with a probability of 0.5 followed by random cropping after zero-padding with size 2 was applied for augmentation.

Table 9: Licenses for datasets used in this work. N-C is used to denote that the data is free for non-commercial use.

| | MNIST | fMNIST | SVHN | CIFAR-10 |
|---|---|---|---|---|
| *License* | CC BY-SA 3.0 | MIT | CC0 1.0 | MIT |

| | STL-10 | QuickDraw | CelebA |
|---|---|---|---|
| *License* | N-C | CC BY 4.0 | N-C |

## A.2 Implementation details

The experiments in this work were conducted on an NVIDIA A100 GPU. The backbone network used for all models as well as all datasets consists of an ImageNet (Deng et al., 2009) pretrained ResNet34 (He et al., 2016). ProtoVAE relies on encoder models from the original work (Gautam et al., 2022) on all datasets but STL-10, QuickDraw, and CelebA, where it is a ResNet34. This choice was made due to the inability of ProtoVAE to converge on certain datasets with a larger encoder like ResNet-34 and a CNN decoder. The size of the latent vector is 512 and the batch size is 128 for all datasets as well as models. Stochastic gradient descent (SGD) is used as the optimizer for training ResNet34 with a momentum of 0.9 for CelebA and 0.5 for all other datasets. For ProtoVAE and FLINT, an Adam (Kingma & Ba, 2015) optimizer is used. Other hyperparameters including learning rate, number of epochs, and number of prototypes are mentioned in Table 10. Note that, unlike other SEMs, KMEx requires tuning of only one additional hyperparameter i.e., the number of prototypes per class, compared to the closest black-box model.

## A.3 Runtimes

Comparing the runtimes for different SEMs is not straightforward owing to the various very specialized and multi-step training schemes. For example, ProtoPNet requires training the whole architecture, followed by the projection of prototypes to the input data, and finally training the last layer for a few more epochs. Therefore, for an exhaustive and complete comparison, we report the full training time for all models in Table 11, where each model is trained to achieve the accuracies reported in Table 3 in the main text, using different hyperparameters, as reported in Table 10. Additionally, we report the test times for a single image for all models in Table 12. These experiments are conducted using PyTorch with a more accessible NVIDIA GeForce GTX 1080 Ti GPU. As can be seen, R34+KMEx requires very low training time as compared to all other SEMs. Further, KMEx is also faster for the prediction of a single test image as compared to other models. These experiments further highlight the effectiveness of the proposed method.

## A.4 Faithfulness of explanations

In this section, we qualitatively evaluate the faithfulness of explanations generated by an SEM to the closest black-box. For this, as mentioned in the main text, we compute Layer-wise Relevance Propagation (LRP) maps (Bach et al., 2015) for prediction probabilities using Zennit (Anders et al., 2021). In Figure 6, we visualize the LRP maps for random images from the CIFAR-10, CelebA, and MNIST datasets. The LRP maps for the black-box ResNet34 and SEMs ProtoPNet, ProtoVAE, and KMEx are shown. For MNIST, we also show LRP maps for a CNN backbone. The CNN architecture used is from (Gautam et al., 2022). As observed, instead of producing non-robust explanations, KMEx remains most faithful to the black-box. This makes KMEx the SEM closest to the corresponding black-box, thereby proving to be an efficient baseline.

## A.5 Detailed Summary plot

We provide the results for the proposed quantitative evaluation framework for each dataset in Figure 7. As observed, KMEx performs the best for transparency, diversity and faithfulness of explanations for all datasets. ProtoVAE performs on par in terms of diversity for all datasets, however it suffers in all other dimensions. ProtoPNet is capable of generating highly robust explanations for nearly all datasets, but it falls short in other areas. FLINT, on the other hand, performs the worst for all evaluations. This detailed summary plot

Table 10: Hyperparameter values for KMEx, ProtoPNet, FLINT and ProtoVAE for all the datasets.

| | | MNIST | fMNIST | SVHN | CIFAR-10 | STL-10 | QuickDraw | CelebA |
|---|---|---|---|---|---|---|---|---|
| *KMEx* | No. of prototypes per class | 5 | 5 | 5 | 5 | 5 | 5 | 20 |
| | No. of epochs | 10 | 10 | 10 | 30 | 30 | 30 | 10 |
| | Learning rate | 0.001 | 0.001 | 0.001 | 0.001 | 0.001 | 0.001 | 0.001 |
| | | | | | | | | |
| *ProtoPNet* | No. of prototypes per class | 5 | 5 | 5 | 5 | 5 | 5 | 20 |
| | No. of epochs | | | | | | | |
| | •warm | 5 | 5 | 5 | 5 | 5 | 5 | 5 |
| | •train | 15 | 15 | 15 | 35 | 35 | 35 | 15 |
| | •push interval | 10 | 10 | 10 | 10 | 10 | 10 | 10 |
| | •last layer | 5 | 5 | 5 | 5 | 5 | 5 | 5 |
| | Learning rates •joint, warm, last layer & prototypes | 0.001 | 0.001 | 0.001 | 0.001 | 0.001 | 0.001 | 0.001 |
| | Loss weights | | | | | | | |
| | •Cross entropy | 1 | 1 | 1 | 1 | 1 | 1 | 1 |
| | •Clustering | 0.8 | 0.8 | 0.8 | 0.8 | 0.8 | 0.8 | 0.8 |
| | •Separation | -0.08 | -0.08 | -0.08 | -0.08 | -0.08 | -0.08 | -0.08 |
| | •$l1$ | 0.004 | 0.004 | 0.004 | 0.004 | 0.004 | 0.004 | 0.004 |
| | | | | | | | | |
| *FLINT* | No. of prototypes per class | 5 | 5 | 5 | 5 | 5 | 5 | 20 |
| | No. of epochs | 10 | 10 | 10 | 30 | 30 | 30 | 10 |
| | Loss weights | | | | | | | |
| | •Cross entropy | 0.8 | 0.8 | 0.8 | 0.8 | 0.8 | 0.8 | 0.8 |
| | •Input fidelity | 0.8 | 0.8 | 0.8 | 0.8 | 0.8 | 0.8 | 0.8 |
| | •Output fidelity | 1.0 | 1.0 | 1.0 | 1.0 | 1.0 | 1.0 | 1.0 |
| | •Conciseness | 0.1 | 0.1 | 0.1 | 0.1 | 0.1 | 0.1 | 0.1 |
| | •Entropy | 0.2 | 0.2 | 0.2 | 0.2 | 0.2 | 0.2 | 0.2 |
| | •Diversity | 0.2 | 0.2 | 0.2 | 0.2 | 0.2 | 0.2 | 0.2 |
| | | | | | | | | |
| *ProtoVAE* | No. of prototypes per class | 5 | 5 | 5 | 5 | 5 | 5 | 20 |
| | No. of epochs | 20 | 20 | 20 | 60 | 60 | 60 | 20 |
| | Loss weights •Cross Entropy | 1 | 1 | 1 | 1 | 1 | 1 | 1 |
| | •Reconstruction | 0.1 | 0.1 | 0.1 | 0.1 | 0.1 | 0.1 | 0.1 |
| | •KL Divergence | 1 | 100 | 100 | 100 | 100 | 100 | 100 |
| | •Orthogonality | 0.1 | 0.1 | 0.1 | 0.1 | 0.1 | 0.1 | 0.1 |

highlights the effectiveness of the proposed qualitative evaluation framework, which can be employed to weigh the advantages and disadvantages of various prototypical SEMs for different datasets.

## A.6 Ablation Study

In this section, we perform ablation studies on the similarity measure used in this work, the number of prototypes for each class, and finally the clustering algorithms.

Table 11: Time (s) for full training

|  | MNIST | fMNIST | SVHN | CIFAR-10 | STL-10 | QuickDraw | CelebA |
|---|---|---|---|---|---|---|---|
| R34+KMEx | **346** | **330** | **348** | **626** | **182** | **662** | **3975** |
| ProtoPNet | 3597 | 3492 | 3006 | 6441 | 733 | 10806 | 20885 |
| FLINT | 429 | 437 | 569 | 1089 | 237 | 1123 | 4489 |
| ProtoVAE | 5030 | 2725 | 6055 | 6313 | 804 | 5694 | 44857 |

Table 12: Single sample test time (s)

|  | MNIST | fMNIST | SVHN | CIFAR-10 | STL-10 | QuickDraw | CelebA |
|---|---|---|---|---|---|---|---|
| R34+KMEx | **0.007** | **0.006** | **0.007** | **0.007** | **0.007** | **0.007** | **0.190** |
| ProtoPNet | 0.011 | 0.204 | 0.018 | 0.072 | 0.011 | 0.011 | 0.210 |
| FLINT | 0.037 | 0.029 | 0.037 | 0.040 | 0.025 | 0.026 | 0.224 |
| ProtoVAE | 0.008 | 0.012 | 0.008 | 0.011 | 0.011 | 0.009 | 0.200 |

CIFAR-10

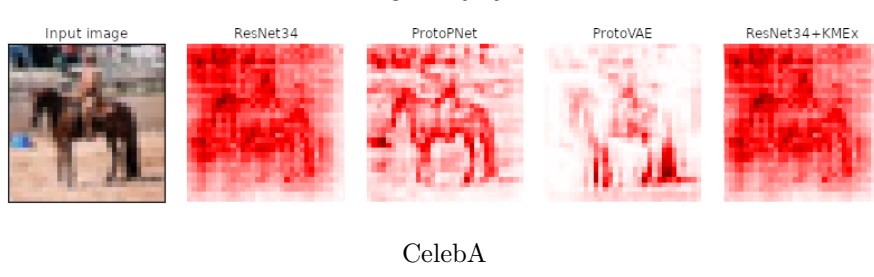

CelebA

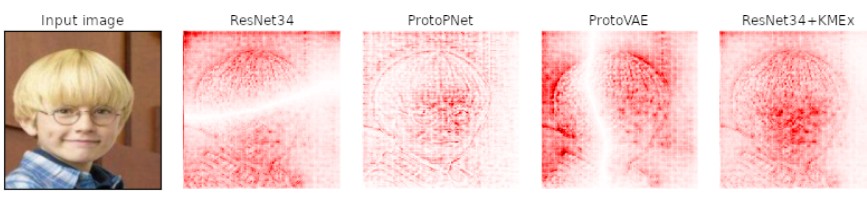

MNIST with a ResNet34 backbone

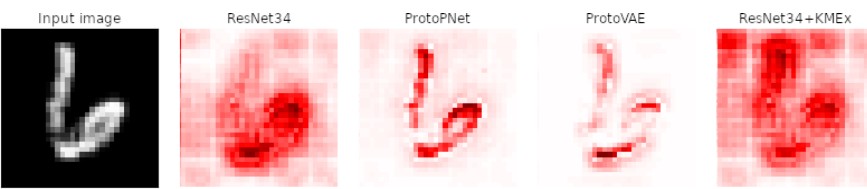

MNIST with a CNN backbone

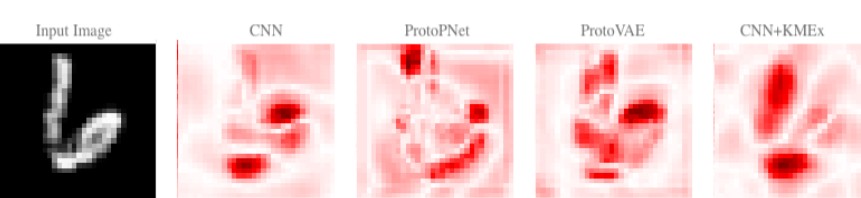

Figure 6: Normalized LRP maps computed on different datasets and with different architectures.

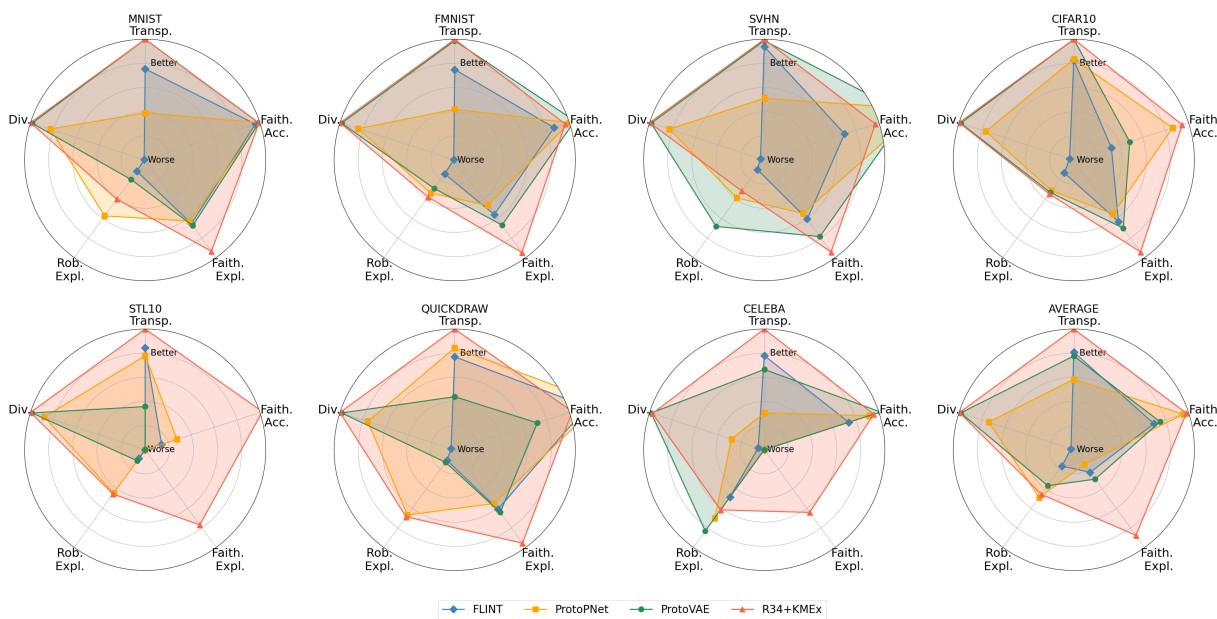

Figure 7: Radar plot summarizing the proposed quantitative evaluation framework for each dataset.

### A.6.1 Similarity measures

In Figure 8, we show the radar plot for the proposed evaluation framework for ablation study on the similarity measure (described in Sec 3.2 of the main text). The similarity measures used for this experiment include $\ell_1$, $\ell_2$, $\log(\ell_2 + 1) - \log(\ell_2 + \epsilon)$, ProtoPNEt's similarity measure $\log((\ell_2^2 + 1)/(\ell_2^2 + \epsilon))$, dot product similarity (dotprod), cosine similarity (cosine) and Normalized Euclidean Distance based similarity (NED). Since downstream operations expect that the more similar, the larger the value, $\ell_1$, $\ell_2$, dotprod, and NED distances are multiplied by $-1$ before further processing.

The first observation is that distance-based measures (except NED) perform all very similarly in terms of transparency, accuracy, and quantitative diversity. This is not surprising since they differ mostly in terms of their tail for larger distances. This seems to be the key to the robustness as both $\ell_1$, $\ell_2$, and dotprod report the best behaviors: these have infinite asymptotes while the others converge to 0. Overall, the $\ell_2$-based similarity performs the best and clearly outperforms all other measures.

### A.6.2 Number of prototypes per class

In the radar plot in Figure 9, we provide full qualitative comparison of KMEx with different numbers of prototypes per class $L$ for all datasets. Note that for $L = 1$ diversity is undefined. In the case of STL10, an increase in the number of prototypes results in a decrease in transparency. This is anticipated as an excessive number of prototypes may result in the capture of redundant information. Apart from this, everything else remains invariant to the number of prototypes. However, from a qualitative perspective, utilizing too few or too many prototypes per class could potentially impact explainability. As depicted in Figure 5, the information captured by the prototypes increases with the increase in prototypes but reaches a saturation point eventually. This experiment further demonstrates the effectiveness of the proposed evaluation framework in selecting hyperparameters, such as the number of prototypes per class, for different models as well as datasets.

### A.6.3 Clustering algorithms

It is indeed possible to utilize other kinds of clustering algorithms instead of K-Means. Yet, this algorithm must be based on centroids in order to have prototypes at the end. Among the centroid-based options, the

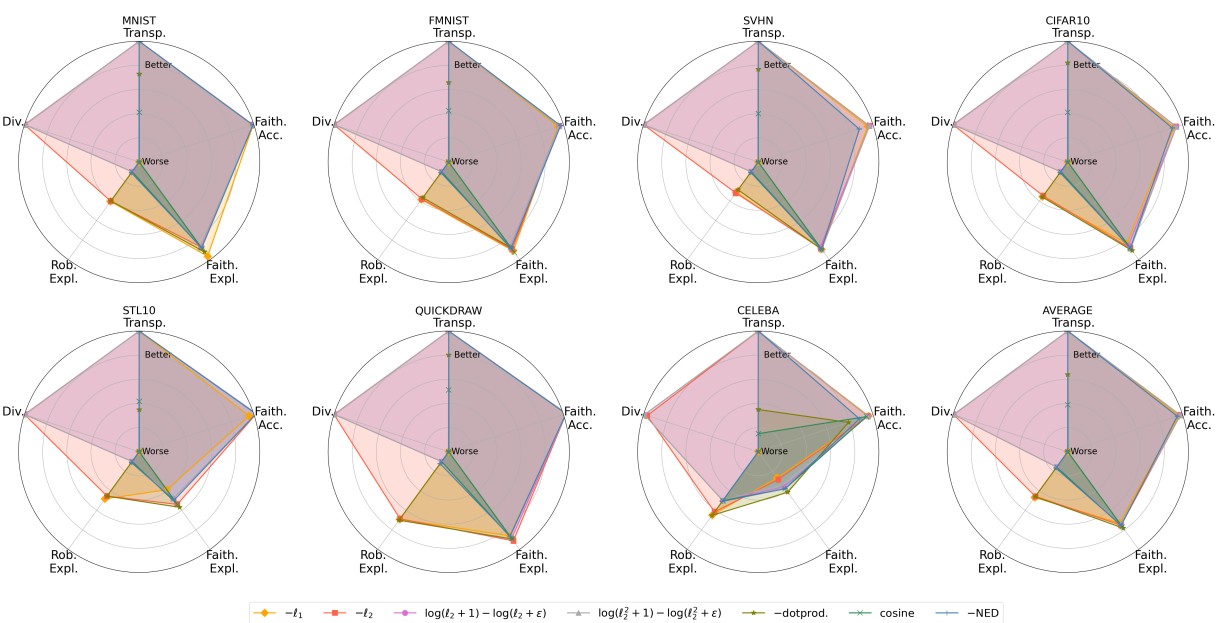

Figure 8: Radar plot for ablation study on the similarity measure.

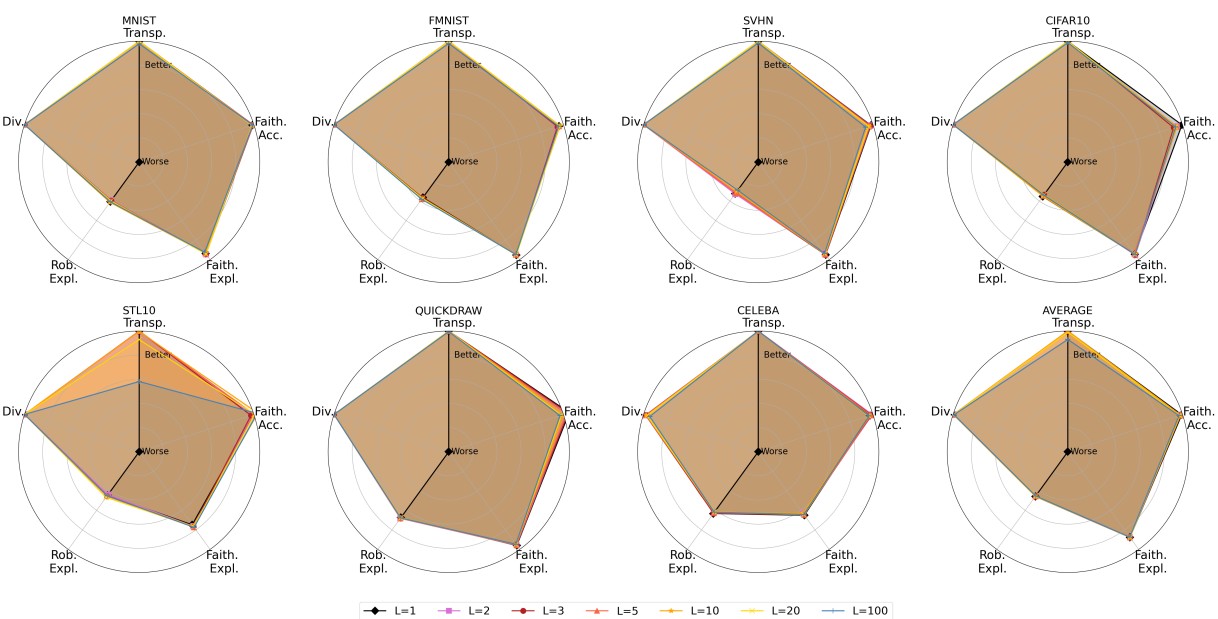

Figure 9: Radar plot for ablation study on the number of prototypes per class.

algorithms are mostly derived from k-means like k-medoids (Park & Jun, 2009) or GMM (gmm, 2019). The drawback of the latter is their high complexity and slow computation time in high dimensions. Bisecting K-Means (Steinbach et al., 2000) is an alternative algorithm that achieves K-Means-like clustering: it splits the data in two using k-means until a convergence criterion is satisfied. However, it still follows a hierarchical structure. Additionally, for a low number of prototypes per class, the centroids of Bisecting K-Means are unlikely to be positioned on accumulation centers of the data as this might unsettle the trade-off between bisecting the clustering and the uniform cluster distribution prior of k-means.

Table 13: Prediction accuracy comparison of R34+KMEx with Bisecting K-Means (R34+BisectingKMEx). Reported numbers are averages over 5 runs along with standard deviations.

|  | MNIST | fMNIST | SVHN | CIFAR-10 | STL-10 | QuickDraw |
|---|---|---|---|---|---|---|
| R34+KMEx | $\mathbf{99.4^{\pm 0.0}}$ | $\mathbf{92.3^{\pm 0.1}}$ | $\mathbf{92.4^{\pm 0.1}}$ | $\mathbf{85.3^{\pm 0.1}}$ | $\mathbf{91.9^{\pm 0.2}}$ | $\mathbf{86.6^{\pm 0.2}}$ |
| R34+BisectingKMEx | $75.1^{\pm 2.6}$ | $62.0^{\pm 6.7}$ | $57.1^{\pm 3.6}$ | $51.1^{\pm 6.0}$ | $58.9^{\pm 4.0}$ | $26.4^{\pm 0.8}$ |

Table 14: Classification performance on CUB200 dataset.

|  | ResNet34 | ResNet34+KMEx | |
|---|---|---|---|
|  |  | Full images | Patches |
| Accuracy | 78.6 | 78.4 | 70.0 |

Since the design of KMEx assumes a non-hierarchical relationship among prototypes, which is in line with all the baselines used in this work, other non-centroid-based alternatives would be inferior to K-Means. To demonstrate this, in table 13, we report the accuracy achieved by R34+BisectingKMEx (where we replace k-means with Bisecting K-Means) and compare it with R34+KMEx. As observed, R34+BisectingKMEx consistently performs worse, with a huge loss in accuracy for all datasets.

### A.7  Preliminary results for a patch-based KMEx on CUB200

We report here preliminary results for the CUB200 (Welinder et al., 2010) dataset. The data consists of 6000 images of 200 classes of birds. We also present a naive extension of KMEx at the patch level. The idea is to compute the patch prototypes right before the final average pool ($7 \times 7 = 49$ patches per image). The class predictions for an image are then derived as the majority vote of the KMEx predictions for each patch.

We report accuracy in percentage in Table 14 for a ResNet34 and its KMEx based on the full images and patches both with 10 prototypes per class. Similarly to (Chen et al., 2019b), we show in Figure 10 the patch prototypes for 10 classes as red rectangles in the closest training image. Note, that some prototypes capture background regions, indicating that the model has learned to exploit background cues.

The drop in accuracy when using patches is not surprising, since the task is more complex. Yet, the results are encouraging and highlight the versatility of KMEx.

### A.8  Additional qualitative results

As mentioned in the main text, quantitative evaluation is not meant to replace the qualitative evaluation of SEMs. Therefore, in this section, we provide qualitative results including prototype visualizations for KMEx. We also show the visual classification strategy used by KMEx using the prototypes for different test examples, thereby exhibiting *this* (image) looks like *that* (prototype). We show this for both correctly and incorrectly test examples to further understand the decision-making process of our SEM. Additionally, we also qualitatively compare the diversity of prototypes for different SEMs.

#### A.8.1  Prototypes learned by KMEx: Figure 11

We visualize the prototypes of KMEx as the images in the training set that have the closest embedding in the latent space to the prototypes. The prototypes are shown for MNIST, fMNIST, SVHN and STL-10 in Figure 11. It can be observed that the prototypes are very diverse and therefore efficiently represent different subgroups of classes.

#### A.8.2  KMEx: *This* looks like *that*: Figure 12

We now visually demonstrate the decision-making process of the proposed SEM. In Figure 12, for random test examples, we show the closest prototype for the MNIST, fMNIST, CelebA, SVHN, STL-10, and CIFAR-10

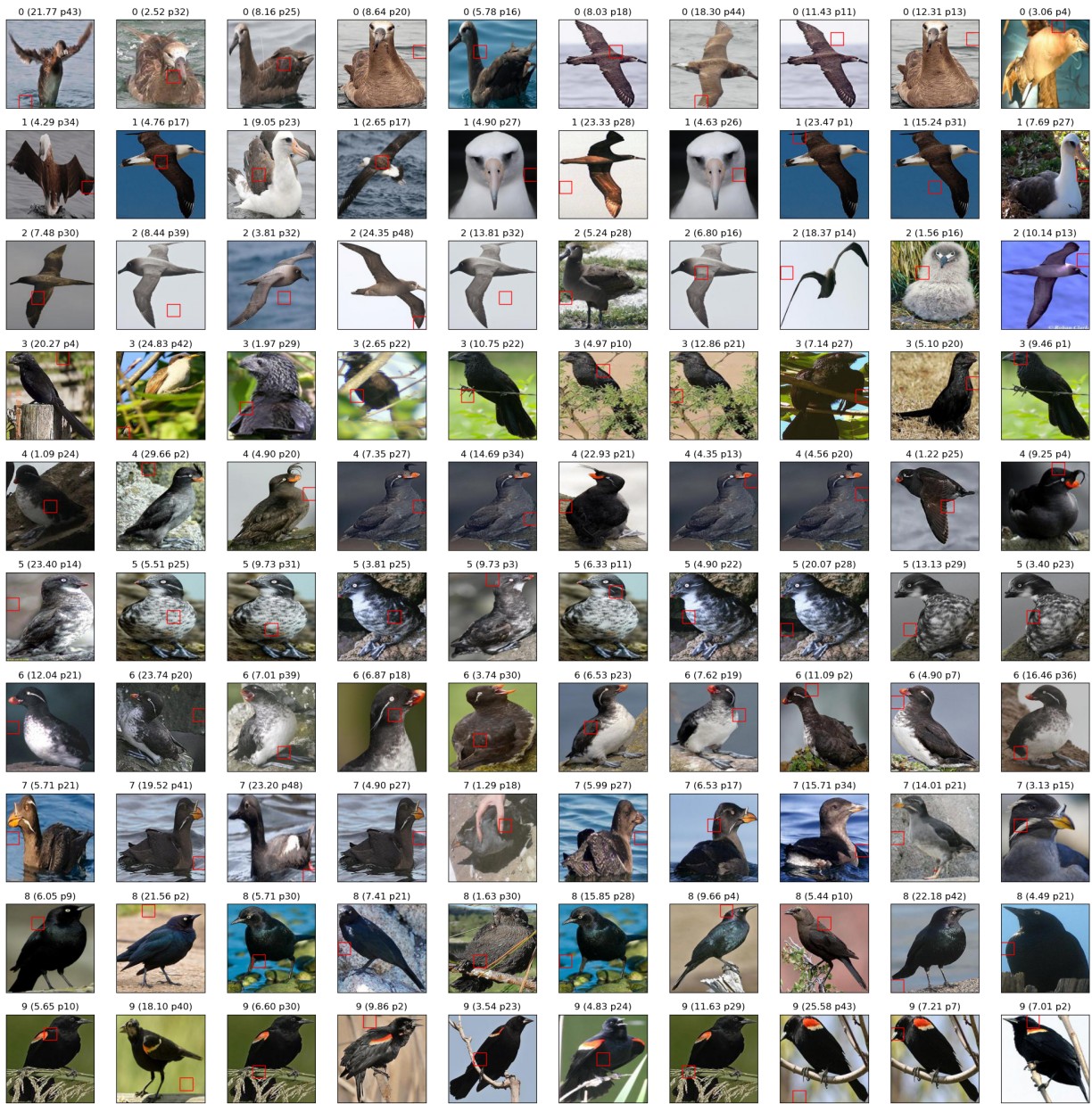

Figure 10: Patch prototypes labeled with class id, importance and patch id.

datasets. It can be observed that the images look very similar to the closest prototypes, which illustrates that representative prototypes are learned. Additionally, we also demonstrate this behavior for misclassified examples (marked by a red rectangle) in Figure 12. As can be seen, the misclassified test images look very similar to prototypes from different classes. Therefore, the simple *this* looks like *that* behavior exhibited by KMEx is able to provide meaningful and transparent decisions.

### A.8.3  KMEx improves the diversity of prototypes: Figures 13-15

We now compare the prototypes of different SEMs, thereby qualitatively comparing the diversity of the prototypes. For consistency and fair comparison, we visualize the closest training images for all the models. In Figure 14, we visualize the prototypes learned by KMEx and ProtoPNet for the MNIST dataset. As

observed, ProtoPNet's prototypes lack diversity, which is especially visible for classes 1, 4, 5, and 7. Applying KMEx on ProtoPNet's embeddings drastically improves the diversity, as shown in Figure 14 (right). Similarly, we compare the prototypes for KMEx and ProtoVAE in Figure 15 for the STL-10 dataset and for KMEx and FLINT in Figure 13. In all the cases, KMEx efficiently improves the diversity of the prototypes.

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

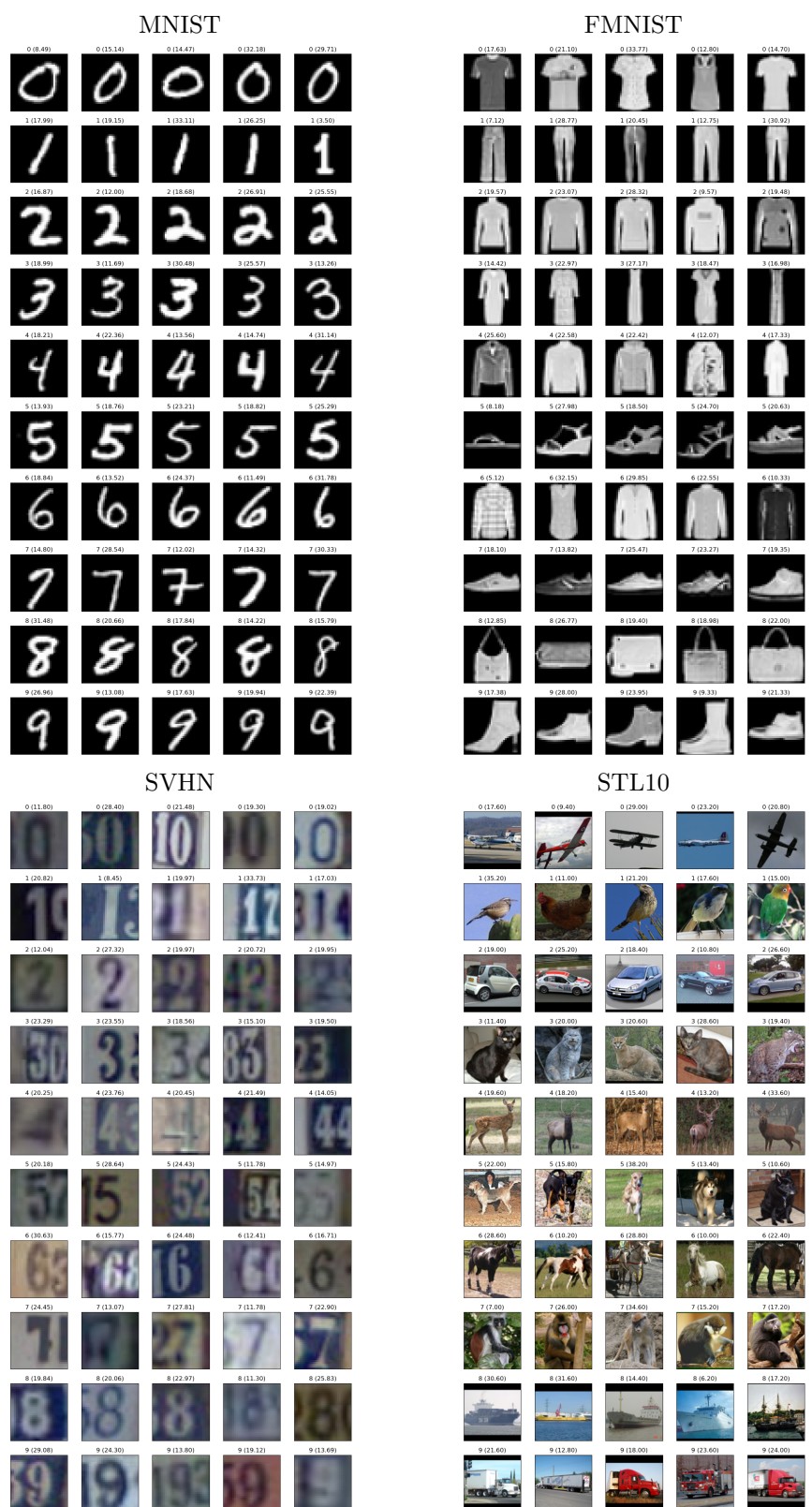

Figure 11: Prototypes learned by KMEx for several datasets. The class label is written on the top of each prototype image along with its importance in the brackets.

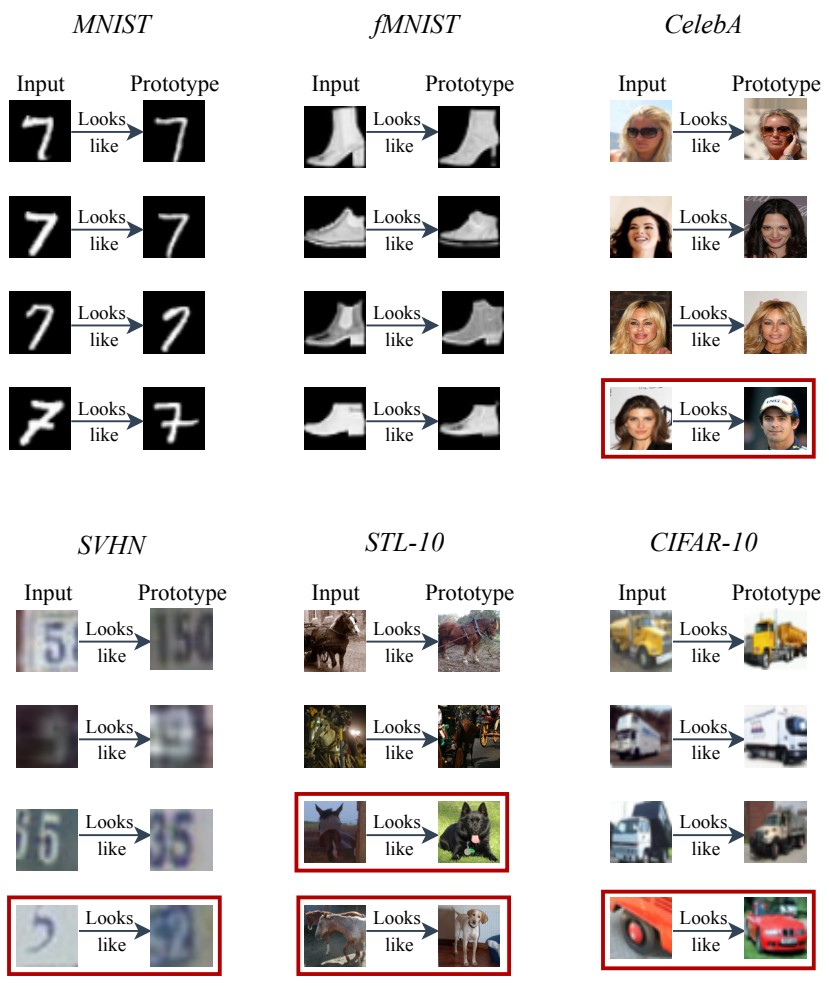

Figure 12: *This* looks like *that* behavior exhibited by KMEx for MNIST, fMNIST, QuickDraw, SVHN, STL-10 and CIFAR-10 datasets. The classification is based on 1-nearest-neighbor, therefore only the closest prototype for each input image is required as the explanation. Misclassified examples are marked in red.

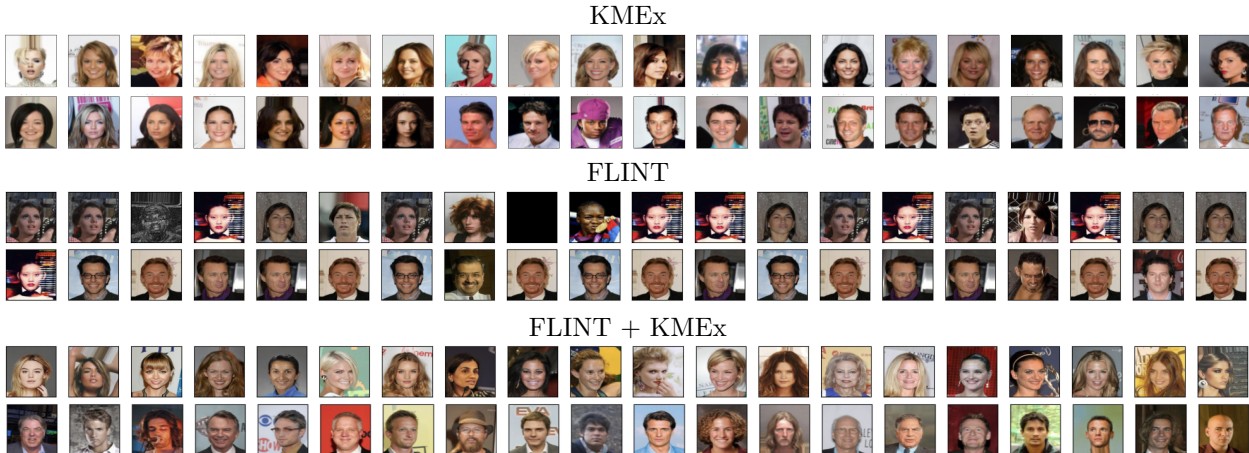

Figure 13: Prototypes learned by KMEx (top) and FLINT (middle) and FLINT-KMEx (bottom) for the CelebA dataset. KMEx generates more diverse prototypes and is again additionally able to improve the prototypes learned over FLINT's embeddings.

KMEx           ProtoPNet           ProtoPNet + KMEx

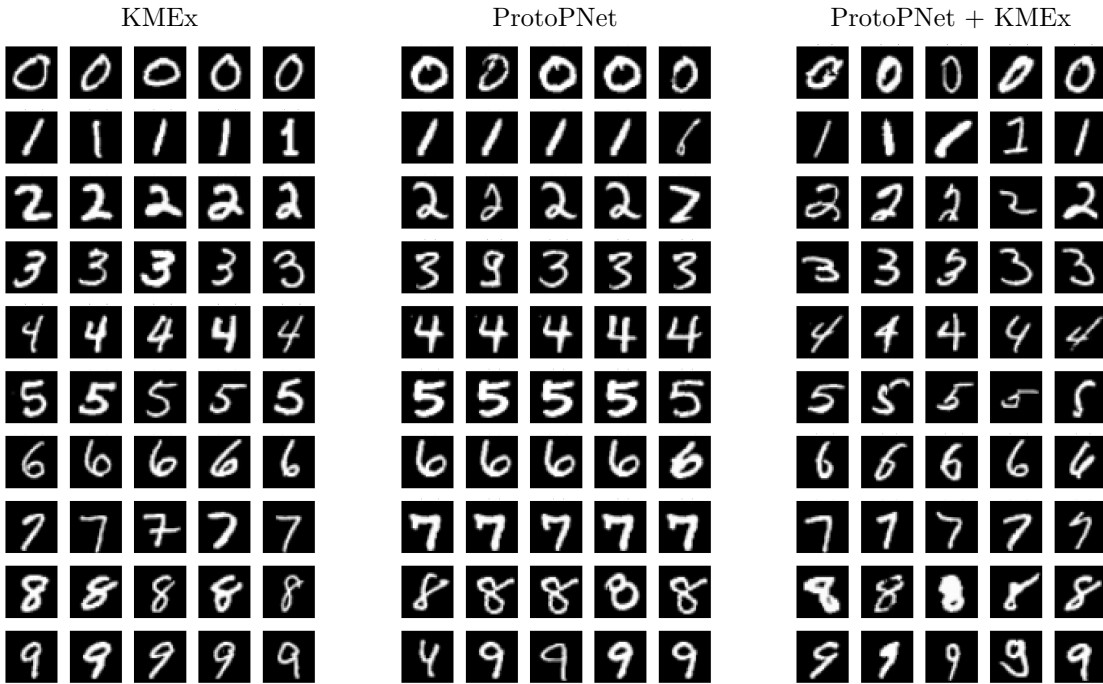

Figure 14: Prototypes learned by KMEx (left) and ProtoPNet (middle) and ProtoPNet-KMEx (right) for the MNIST dataset. KMEx generates more diverse prototypes and is additionally able to improve the prototypes learned over ProtoPNet's embeddings.

KMEx           ProtoVAE           ProtoVAE + KMEx

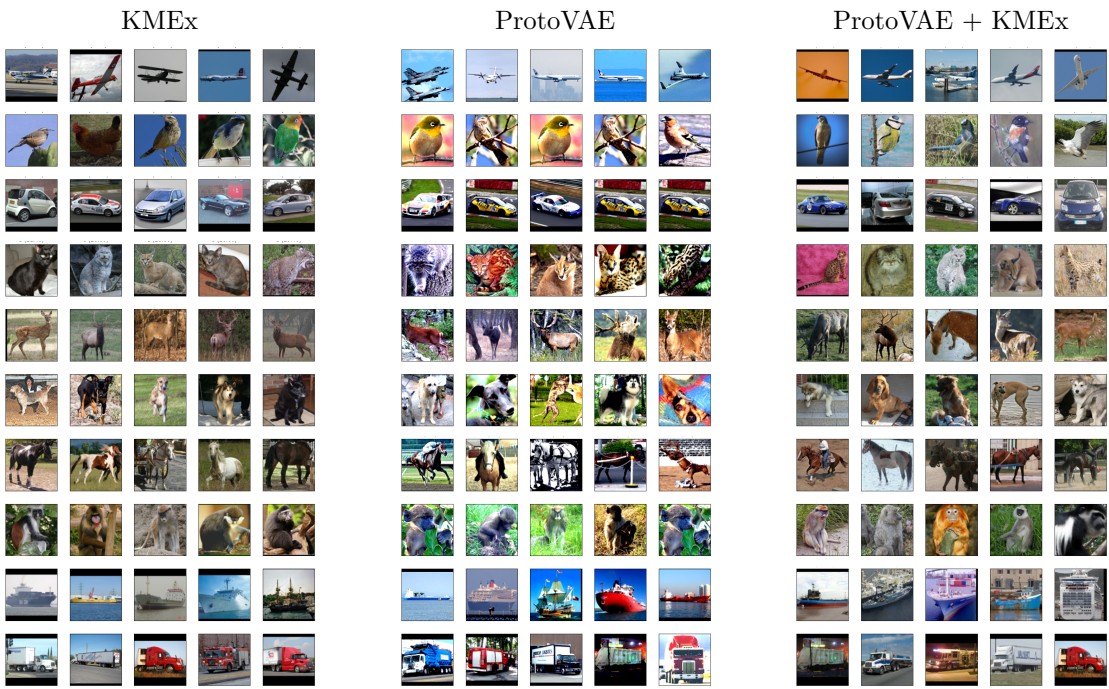

Figure 15: Prototypes learned by KMEx (left) and ProtoVAE (middle) and ProtoVAE-KMEx (right) for the STL-10 dataset. KMEx generates more diverse prototypes and is again additionally able to improve the prototypes learned over ProtoVAE's embeddings.

