# OpenReview forum: "Prototypical Self-Explainable Models Without Re-training"
_TMLR — Accepted by TMLR_

### Review · Reviewer_GmHw · 2024-01-15

**Summary Of Contributions:**

The paper presents KMEx (K-Means Explainer) a post-hoc XAI technique to convert pre-trained models into prototypical interpetable-by-desing distance-based models. KMEx is compared against state-of-the-art prototypical-based approaches through an objective evaluation to assess predictive performance, the robustness of explanations, transparency, concept ghosting, faithfulness, and diversity of the explanations.

**Audience:**

Yes

**Claims And Evidence:**

Yes

**Requested Changes:**

The idea behind KME is simple and intuitive. This makes the approach widely usable and effective. We recommend the authors fix the following issues/missing aspects in order to improve the soundness of the paper.

First, there is no theoretical nor empirical justification of why the similarity measure on page 4 is selected/designed. Both aspects should be curated, analyzing the sensitivity of  KME to alternative similarity measures such as  1-NormalizedEuclidean or other baselines based on existing metrics.

Second, a crucial aspect of the proposal is its usability. However, runtimes are not reported while I believe that these would further highlight the effectiveness of the proposal.

Third, being based on k-Means I expected a study on the sensitivity of the evaluation metrics reported w.r.t. the number of clusters k. Besides, theoretically, any clustering algorithm can be used, but it is better if a centroid-based one is selected in order to rely on prototypes by construction. However, I would love to see that, indeed, k-Means is the best choice and not some other alternatives such as bisective k-Means that would also provide a hierarchy or even directly use a Hierarchical Agglomerative clustering with Ward criterion.

Finally, the statistical significance of the experiments should be reported to show if the various alternatives are statistically exchangeable or f KME is a method that is significantly better than the others.

**Strengths And Weaknesses:**

+ Simple idea
+ Variegated experiments and evaluation measures

- Lack of runtime
- Lack of deeper experiments on clustering
- Lack of statistical significance

---

> ### Author Response · Authors · 2024-02-22
> **Response to Reviewer GmHw Part 1**
>
> Thank you for the thorough review and for acknowledging the effectiveness of the proposed methodology. We would like to thank the reviewer for identifying the aspects in our paper that can be improved, and are delighted to address them in the following:
>
> - **Statistical significance of the experiments**: the statistical significance of the experiments should be reported to show if the various alternatives are statistically exchangeable.
> - **Ablation on similarity measure**: There is no theoretical nor empirical justification of why the similarity measure on page 4 is selected/designed.
> - **Ablation on number of clusters k**: a study on the sensitivity of the evaluation metrics reported w.r.t. the number of clusters k
> - **Ablation on clustering algorithms**: k-Means is the best choice and not some other alternatives such as bisective k-Means that would also provide a hierarchy.
> - **Comparison of runtimes**: runtimes are not reported
>
> Before answering these points separately, we would like to inform the reviewer of two major changes.
>
> 1. We discovered that PyTorch's ``cdist`` function, in its default setting, computes an approximation of the true distance between two vectors. These undesired approximations have slight repercussions on the scores that we compute except for the accuracy, as the ranking of the similarity scores on which the prediction is based is still preserved. We have therefore re-computed the scores on the previously trained models using the following explicit formula: `` torch.sqrt(torch.square(ebd[:,None,:] - prt[None,:,:]).sum(-1)) ``.
> While results differ slightly, the essence of the message does not change: KMEx offers a quick and efficient way to explain an opaque classifier using prototypes.
>
>
> 2. Following the ablation study suggested by reviewer GmHw, we changed the similarity measure used in KMEx to the negative $\ell_2$ distance. This change as well does not alter the essence of the contribution, yet it makes our model more robust and stable. This is discussed in the **Ablation on similarity measure** section of this answer, as well as added to the Appendix in Sec. A.6.1 of the revised manuscript. In addition to the change in similarity score, we now decided to report the area under the Relevance Ordering (RO) curve with the least relevant pixels removed first in Table 4, instead of AI and AD scores.
> As the RO test is based on class probabilities, it is robust to the scaling and shape of the embedding, thus making it a better choice for comparing the different models. The detailed results are discussed in Sec. 4.2.2 of the main text.
>
> The above changes have led to changes in Tables 5, 6 and 7. Nonetheless, again, the main idea of KMEx remains intact and has, in fact, become more robust. We have further added the missing results for FLINT in Table 6. As observed in Table 6, with the fixed distance implementation, R34+KMEx returns, as expected, the lowest scores for the faithfulness of the explanations.
> The corrected Table 7 highlights the clear advantage of KMEx in terms of computed diversity.
> This advantage can also be seen in Table 8 where for FLINT's embeddings more diverse prototypes are learned using KMEx. However the same is not reflected for ProtoPNet or ProtoVAE's embeddings.
>
> ### **Statistical significance of the experiments**
> We have updated and highlighted in bold the statistically significant best results for Tables 3 to 7.
> We relied on Scipy's ``scipy.stats.ttest_ind`` with standard parameters (two-sided).
> It is now easier to see in Table 3 that R34+KMEx achieves the best accuracies for 3 out of 7 datasets. Interestingly, none of the compared SEMs is able to outperform the black-box model on CIFAR-10.
> In Table 5, some models achieve non-zero transparency scores without being statistically different from a plain 0 with a null standard deviation. Nonetheless, the latter is the desired transparency score.

---

> > ### Author Response · Authors · 2024-02-22
> > **Response to Reviewer GmHw Part 2**
> >
> > ### **Ablation on similarity measure**
> > Since we built our methodology on ProtoPNet and ProtoVAE, we directly adopted the same similarity measure used in both of these works.
> >
> > However, following the reviewer's suggestion, we added an ablation study on the similarity measure. In Figure 8 in the Appendix, we have added a radar plot based on our proposed evaluation framework used in this work. The similarity measures used for this experiment include  $\ell_1$, $\ell_2$, $\log{\ell_2+1}-\log(\ell_2+\epsilon)$ our original similarity measure $\log((\ell_2^2+1)/(\ell_2^2+\epsilon))$, dot product similarity (dotprod), cosine similarity (cosine), and Normalized Euclidean Distance based similarity (NED). Since downstream operations expect that the more similar, the larger the value,  $\ell_1$, $\ell_2$, dotprod, and NED distances are multiplied by $-1$ before further processing.
> >
> > The first observation is that distance-based measures (except NED) perform all very similarly in terms of transparency, accuracy, and quantitative diversity. This is not surprising since they differ mostly in terms of their tail for larger distances. This seems to be the key to the robustness as both $\ell_1$, $\ell_2$, and $\operatorname{dot-prod}$ report the best behaviors: these have infinite asymptotes while the others converge to $0$.
> >
> > Overall, the $\ell_2$-based similarity performs the best and clearly outperforms our originally used measure in terms of the robustness of explanations.
> > We have therefore adopted the $\ell_2$ as the similarity measure for KMEx and changed the main text accordingly:
> >
> > 1. Change in the similarity measure in Section 3.2.
> > 2. Changes in the results in Tables 5-7.
> > 3. Change in radar plot in Figure 4.
> >
> > Again, we would like to thank the reviewer for the suggestion which helped us to improve our model.
> >
> > ### **Ablation on number of clusters**
> > In Section 4.4 of the paper, we conducted a study on the CelebA dataset to examine the correlation between the increase in the number of prototypes and the amount of information captured.
> > However, we agree that a comprehensive evaluation for all datasets with respect to the number of prototypes is missing from the paper. We deliberately disregarded this in the original manuscript, considering the focus on comparison with other SEM baselines, where the same number of clusters/prototypes is being used for all other SEMs in our work to maintain consistency in comparisons. However, we agree that this experiment can further demonstrate the importance of the number of clusters in the proposed KMEx. In the radar plot in Figure 9 of the Appendix, we provide a full qualitative comparison of KMEx with different $L$ (prototypes per class) for all datasets. Note that for $L=1$ diversity is undefined. In the case of STL10, an increase in the number of prototypes results in a decrease in transparency. This is anticipated as an excessive number of prototypes may result in the capture of redundant information. Apart from this, everything else remains invariant to the number of prototypes. However, from a qualitative perspective, utilizing too few or too many prototypes per class could potentially impact explainability. Further, as depicted in Figure 5, the information captured by the prototypes increases with the increase in prototypes but reaches a saturation point eventually. This experiment further demonstrates the effectiveness of the proposed evaluation framework in selecting hyperparameters, such as the number of prototypes per class, for different models as well as datasets.

---

> > > ### Author Response · Authors · 2024-02-22
> > > **Response to Reviewer GmHw Part 3**
> > >
> > > ### **Ablation on clustering algorithms**
> > > It is indeed possible to build on our idea and convert a black-box model using other kinds of clustering algorithms. Yet, as rightfully noted by the reviewer, this algorithm must be based on centroids in order to have prototypes at the end.
> > > Therefore, a Hierarchical Agglomerative clustering (HAC) with the Ward criterion does not fit since it does not provide centroids per construction. Another point is that HAC is not designed for the prediction of new data points, therefore it is unclear how to use it in a classifier.
> > >
> > > The remaining centroid-based options are mostly derived from k-means like k-medoids or GMMs. The drawback of the latter is their high complexity and slow computation time in high dimensions.
> > > Bisecting-k-means is an alternative algorithm that achieves k-means-like clustering: it splits the data in two using k-means until a convergence criterion is satisfied. However, it still follows a hierarchical structure. Additionally, for a low number of prototypes per class, the centroids of Bisecting-k-means are unlikely to be positioned on accumulation centers of the data as this might contradict the uniform cluster distribution prior to k-means.
> > >
> > > Since the design of KMEx assumes a non-hierarchical relationship among prototypes, which is in line with all the baselines used in this work, we agree with the reviewer that other non-centroid-based alternatives would be inferior to k-means.
> > > However, to demonstrate this, in the following table, we report the accuracy achieved by R34+BisectingKMEx (where we replace k-means with Bisecting k-means) and compare it with R34+KMEx.
> > > As observed, R34+BisectingKMEx consistently performs worse, with a huge loss in accuracy for all datasets. We have further added this discussion in Sec. A.6.3 in the Appendix.
> > >
> > > || MNIST      |       fMNIST    |       SVHN    |        CIFAR-10    |      STL-10     |        QuickDraw |
> > > |---|---|---|---|---|---|---|
> > > |R34+KMEx| $\boldsymbol{99.4 ^ {\pm 0.0}}$ | $\boldsymbol{92.3 ^ {\pm 0.1}}$ | $\boldsymbol{92.4 ^ {\pm 0.1}}$ | $\boldsymbol{85.3 ^ {\pm 0.1}}$ | $\boldsymbol{91.9 ^ {\pm 0.2}}$ | $\boldsymbol{86.6 ^ {\pm 0.2}}$ |
> > > |R34+BisectingKMEx|  $ 75.1 ^ {\pm 2.6}$ | $ 62.0 ^ {\pm 6.7}$ | $ 57.1 ^ {\pm 3.6}$ | $51.1 ^ {\pm 6.0}$ | $58.9 ^ {\pm 4.0}$ | $26.4 ^ {\pm 0.8}$ |
> > >
> > >
> > > ### **Comparison of runtimes**
> > > We thank the reviewer for recognizing the usability of the approach with respect to runtimes. Comparing the runtimes for different SEMs is not straightforward owing to the various very specialized and multi-step training schemes.
> > > Therefore, for an exhaustive and complete comparison, we report the full training time for all models, where each model is trained to achieve the accuracies reported in Table 3 in the main text, using different hyperparameters, as reported in Table 10 in the Appendix.
> > > Additionally, we report the test times for a single image for all models.
> > > Our proposed R34+KMEx is the fastest to train and compute predictions.
> > > These experiments further highlight the effectiveness of the proposed method, as suggested by the reviewer. We have added this discussion in Sec. A.3 in the Appendix.
> > >
> > > Time (s) for full training
> > > || MNIST            | fMNIST           | SVHN             | CIFAR-10          | STL-10            | QuickDraw  | CelebA|
> > > |---|---|---|---|---|---|---|---|
> > > |R34+KMEx | $ \boldsymbol{346} $ | $ \boldsymbol{330} $ | $ \boldsymbol{348} $ | $ \boldsymbol{626} $ | $ \boldsymbol{182} $ | $ \boldsymbol{662} $ | $ \boldsymbol{3975} $|
> > > |ProtoPNet| $ 3597 $ | $ 3492 $| $ 3006 $ | $ 6441 $ | $ 733 $ | $ 10806 $ | $ 20885 $ |
> > > |FLINT| $ 429 $ | $ 437 $ | $ 569 $ | $ 1089 $ | $ 237 $ | $ 1123 $ | $ 4489 $ |
> > > |ProtoVAE| $ 5030 $ | $ 2725 $ | $ 6055 $ | $ 6313 $ | $ 804 $ | $ 5694 $ | $ 44857 $ |
> > >
> > > Single sample test time (s)
> > > || MNIST            | fMNIST           | SVHN             | CIFAR-10          | STL-10            | QuickDraw  | CelebA|
> > > |---|---|---|---|---|---|---|---|
> > > |R34+KMEx| $ \boldsymbol{0.007} $ |$ \boldsymbol{0.006} $ | $ \boldsymbol{0.007} $ | $ \boldsymbol{0.007} $ | $ \boldsymbol{0.007} $ | $ \boldsymbol{0.007} $ | $ \boldsymbol{0.190} $ |
> > > |ProtoPNet| $ 0.011 $ | $ 0.204 $ | $ 0.018 $ | $ 0.072 $ | $ 0.011 $ | $ 0.011 $ | $ 0.210 $ |
> > > |FLINT | $ 0.037 $ | $0.029 $ | $ 0.037 $ | $ 0.040 $ | $ 0.025 $ | $ 0.026 $ | $ 0.224 $ |
> > > |ProtoVAE | $ 0.008 $ | $ 0.012$ | $ 0.008 $ | $ 0.011 $ | $ 0.011 $ | $ 0.009 $ | $ 0.200 $ |

---

### Review · Reviewer_J8fa · 2024-02-09

**Summary Of Contributions:**

- The authors proposed a novel self-explainable model called KMEx.
- The advantages of KMEx is that we can apply on top of most black-box models only with limited amount of computations (k-mean computations).
- The authors also proposed some interesting metrics for SEM. Note that evaluation of interpretability is challenging.
- The authors provided various experiments to show the pros and cons of the proposed method in comparison to the baselines.

**Audience:**

Yes

**Broader Impact Concerns:**

In general, this is XAI paper. Thus, it would be important to clarify that explanations that the proposed method is provided are not perfect and should not fully believe the explanation.

**Claims And Evidence:**

No

**Requested Changes:**

**1. SEMs on safety-critical scenarios**
- Could we really utilize the proposed SEMs for safety-critical scenarios?
- As explained, SEMs are preferred for safety-critical scenarios. However, it is on the representations of black-box models and those black-box models are totally un-interpretable. In that case, only with the interpretable components after representation extraction, could we use this method for "safety-critical" scenarios?
- If the representation is not perfect, all the following explanations inherit the issues of the representations and we do not know the issues of the representation interpretably. So, I think it is very hard to utilize the proposed SEMs on "safety-critical" scenarios
- Also, if this approach can be used for these scenarios, I think all the other post-hoc XAI approaches can be also utilized for the same scenarios.
- The statement that KMEx is not a post-hoc method (in Section 3) is weak. Please explain more based on my above comments.

**2. Difference from other prototype based XAI**
- I think there are many prototype based XAI including this paper (https://papers.nips.cc/paper_files/paper/2016/file/5680522b8e2bb01943234bce7bf84534-Paper.pdf) and the papers in the related works of this paper.
- It is highly unclear how this proposed method is different from these works.
- It would be good to add more thorough related works about comparing the proposed method with other prototype based XAI.

**3. Method section needs more details**
- In Fig 1, what is D_tsp? It seems like the explanation comes in Section 4 which is way far from Fig. 1.
- Also, based on Section 4, with KMEx, D_tsp = 0.0. In that case, 1-D_tsp = 1 for all? But why are the sizes of squares different?
- Do we use the same "k" values per class?
- What is z? What is p_k? What is epsilon?
- It would be always good to define the notation first before using them in the equation.
- It seems like the authors claim that trustworthiness of the explanation comes from the black-box model (this part: ...to both, thus similar explanations...). But we cannot say that the black-box model's embedding space is trustworthy.

**4 Experiments (until Section 4.1)**
- Table 3. Could you explain why some baselines have better results than the proposed method and even baseline method? Can this be the disadvantage of the proposed method?
- Figure 2. Could we show the qualitative results of other methods that we can compare with? Also good to add how we can interpret PRP (like what is the meaning of the red part)
- Table 4. It seems that the performance of KMEx is significantly and consistently worse than ProtoPNet. So, in that case, could we understand that KMEx is less robust? In that case, could we use this method for "safety-critical" scenarios where robustness is key?

**5. Experiments (Section 4.2)**
- Notation part should come much earlier (like Section 3)- Good to define D as well.
- Good to provide the example of "s"
- Equation (1). It seems like Equation (1) is wrong. See the inside of sigma. We can take out 1/KN part and now we can focus on the nominator with the sigma. As we know, each sample has one nearest prototype. Thus, the sum of the nominator should be N. In that case, D_tsp = 1/K regardless of the method. I do not think this is the definition that the authors want to define.
- Also, I think using "testing set" instead of "training set" to evaluate D_tsp would be more meaningful because we would like to use this XAI in a test scenario. Working well with training data may not impact a lot.
- What is p. in Equation (2)?
- In Table 7, ProtoVAE is much better than KMEx. Is this the right understanding?

**Strengths And Weaknesses:**

**Strength:**
- The proposed method is simple but efficient to provide explanation on top of most black-box models.
- Some of the proposed evaluation metrics make sense and promising.

**Weakness:**
- Methodology section needs much more details. Currently, most details are missing or wrongly located.
- It is unclear how the proposed SEMs can be used in "safety-critical" scenarios given that it is implemented on top of black-box models.
- It would be good to clarify more that how the proposed method is different from post-hoc explanation methods with prototypes.
- In multiple metrics, the proposed method shows worse performance. It is unclear why we should use the proposed method even though we see these worse performance.
- More can be found in the requested changes section.

---

> ### Author Response · Authors · 2024-02-22
> **Response to Reviewer J8fa Part 1**
>
> We thank the reviewer for recognizing the simplicity and efficiency of the proposed SEM, KMEx, as well as the proposed evaluation framework.
>
> We agree that some details can be clarified or their presentation can be moved to further improve the clarity of the paper. Along with the necessary changes for adding more details in the methodology section, we have addressed your concerns below.
>
> Before all, we would like to inform the reviewer of two major changes.
>
> 1. We discovered that PyTorch's ``cdist`` function, in its default setting, computes an approximation of the true distance between two vectors. These undesired approximations have slight repercussions on the scores that we compute except for the accuracy, as the ranking of the similarity scores on which the prediction is based is still preserved. We have therefore re-computed the scores on the previously trained models using the following explicit formula: `` torch.sqrt(torch.square(ebd[:,None,:] - prt[None,:,:]).sum(-1)) ``.
> While results differ slightly, the essence of the message does not change: KMEx offers a quick and efficient way to explain an opaque classifier using prototypes.
>
>
> 2. Following the ablation study suggested by reviewer GmHw, we changed the similarity measure used in KMEx to the negative $\ell_2$ distance. This change as well does not alter the essence of the contribution, yet it makes our model more robust and stable. This discussion is added to the Appendix in Sec. A.6.1 of the revised manuscript. In addition to the change in similarity score, we now decided to report the area under the Relevance Ordering (RO) curve with the least relevant pixels removed first in Table 4, instead of AI and AD scores.
> As the RO test is based on class probabilities, it is robust to the scaling and shape of the embedding, thus making it a better choice for comparing the different models. The detailed results are discussed in Sec. 4.2.2 of the main text.
>
> The above changes have led to changes in Tables 5, 6 and 7. Nonetheless, again, the main idea of KMEx remains intact and has, in fact, become more robust. We have further added the missing results for FLINT in Table 6. As observed in Table 6, with the fixed distance implementation, R34+KMEx returns, as expected, the lowest scores for the faithfulness of the explanations.
> The corrected Table 7 highlights the clear advantage of KMEx in terms of computed diversity.
> This advantage can also be seen in Table 8 where for FLINT's embeddings more diverse prototypes are learned using KMEx. However the same is not reflected for ProtoPNet or ProtoVAE's embeddings.

---

> ### Author Response · Authors · 2024-02-22
> **Response to Reviewer J8fa Part 2**
>
> **1. SEMs in safety-critical scenarios**
>
>  > - Could we really utilize the proposed SEMs for safety-critical scenarios?
> > - As explained, SEMs are preferred for safety-critical scenarios. However, it is on the representations of black-box models and those black-box models are totally un-interpretable. In that case, only with the interpretable components after representation extraction, could we use this method for "safety-critical" scenarios?
> > - If the representation is not perfect, all the following explanations inherit the issues of the representations and we do not know the issues of the representation interpretably. So, I think it is very hard to utilize the proposed SEMs on "safety-critical" scenarios
>
> The reviewer raised the point that the embedding learned by the original black-box is not interpretable. However, we would like to argue that even for ProtoPNet's or any model's encoder with, let's say, more than two layers the embedding is non-interpretable. The difference in prototypical SEMs lies in the transparency of the final predictions, which are based on the interpretable prototypical layer, followed by an interpretable classifier.
> Several authors have also made this claim or investigated the fitness of ProtoPNet's paradigm, which we share, for so-called "safety-critical" scenarios [1, 2, 3]. Additionally, we would like to state that the aim of these prototypical SEMs is not to learn perfect representations, but instead to make the learned representation space globally interpretable through prototypes. As an example, [4] shows the importance of using ProtoPNet in a safety-critical scenario where the data, as well as, the learned representations are not perfect.
>
> [1] Eunji Kim, Siwon Kim, Minji Seo, and Sungroh Yoon. Xprotonet: Diagnosis in chest radiography with global and local explanations. In Proceedings of the IEEE/CVF Conference on Computer Vision and Pattern Recognition (CVPR), pages 15719–15728, June 2021.
>
> [2] Carloni, Gianluca, et al. "On the applicability of prototypical part learning in medical images: breast masses classification using ProtoPNet." International Conference on Pattern Recognition. Cham: Springer Nature Switzerland, 2022.
>
> [3] Mohammadjafari, Sanaz, et al. "Using ProtoPNet for Interpretable Alzheimer's Disease Classification." Canadian Conference on AI. 2021.
>
> [4] Srishti Gautam, Marina M.-C. Höhne, Stine Hansen, Robert Jenssen, and Michael Kampffmeyer. Demonstrating the risk of imbalanced datasets in chest x-ray image-based diagnostics by prototypical relevance propagation. In 2022 IEEE 19th International Symposium on Biomedical Imaging (ISBI), pp. 1–5, 2022b.
>
>
> > - Also, if this approach can be used for these scenarios, I think all the other post-hoc XAI approaches can be also utilized for the same scenarios.
>
> XAI methods aim for the same objective: explaining a prediction in an intelligible manner without digging into the weights [5].
> The applicability to safety-critical scenarios depends on the robustness and consistency of their predictions and explanations. For KMEx and its global explanations, these two criteria depend on the encoder and k-means, respectively. Luckily, there exist several models that are designed for robustness to adversarial attacks or overfitting than a plain feedforward classifier, and these could also be combined with KMEx.
>
> To go a little further, there is one mathematically certain fact about the 'usual' deep encoders: two points close in the embedding share features in the input space that the network recognizes as similar.
> They acquire this property for being a composition of continuous functions.
> Prototypical and concept-based models build on this property as they give tools to understand the encoder through the embedding it produces. KMEx goes one step further as it replaces linear classifiers with a knn classifier, which allows tracing back a prediction to the proximity of a data point with a prototype. A feat that is not obvious for a linear classifier.
> Moreover, the use of k-means ensures that the prototypes are unlikely out-of-distribution since the algorithm rather positions them on points of accumulation of the embedding (if they exist), as this would minimize its objective function. Such accumulation points indicate that the encoder attributed a lot of attention to some specific features shared by the training data. Such insight is, for example, not accessible to saliency-based XAI methods.
>
> [5] Aytekin, Caglar. "Neural networks are decision trees." arXiv preprint arXiv:2210.05189, 2022.

---

> > ### Author Response · Authors · 2024-02-22
> > **Response to Reviewer J8fa Part 3**
> >
> > > -  The statement that KMEx is not a post-hoc method (in Section 3) is weak. Please explain more based on my above comments.
> >
> > We would like to thank the reviewer for bringing to light that our explanations were not sufficient. We considered further this point and have softened our claim. Now it reads:
> >
> >  "Note that the KMEx conversion is not a post-hoc explainability method per se as what it produces is another model that is inherently interpretable thanks to the central role of the prototypes and the transparent classifier.
> >  Although the trained encoder is re-used, the KMEx's predictions are computed differently using prototypes that were not part of the initial model."
> >
> > **2. Difference from other prototype based XAI**
> >
> > >- I think there are many prototype based XAI including this paper (https://papers.nips.cc/paper_files/paper/2016/file/5680522b8e2bb01943234bce7bf84534-Paper.pdf) and the papers in the related works of this paper.
> > >- It is highly unclear how this proposed method is different from these works.
> > >- It would be good to add more thorough related works about comparing the proposed method with other prototype based XAI.
> >
> > This work of Been Kim et al. (MMD-critic) also looks for prototypes, and counter prototypes, aka "criticisms", in a trained embedding. The difference lies in the method they use to learn them (MMD-based loss with regularizer) and the usage they make of the prototypes i.e, to improve the interpretability of complex data distributions, however not necessarily using them for machine learning based predictions. In our case, we use a simple k-means to obtain prototypes and use them to make predictions. Yet, in both cases, the prototypes serve as global explanations of a class. While arguably MMD-critic could be used instead of k-means to select prototypes, the underlying assumptions for this new model would be completely different from KMEx as well as other prototypical SEMs considered in this work, due to the presence of "criticisms" and is therefore out of scope of this work. Additionally, we would like to state that the main aim of this work is to contribute to the research for novel prototypical SEMs (difference with which is discussed below) and not to other prototypical XAI. As stated earlier, the difference in prototypical SEMs, as opposed to other XAI methods, lies in the transparency of the final predictions in a deep learning model, which are based on the interpretable prototypical layer, followed by an interpretable classifier. This is the essence of KMEx as well.
> >
> > One main difference between KMEx and other SEMs, which aim to be explainable alternatives to a black-box classifier, is that the training of KMEx's backbone does not depend on the prototypes.
> > A risk with SEMs involving the prototypes during the learning of the embedding is that they may create artificial points of accumulation. The global explanations may capture more diversity of the data but not necessarily one relevant for the classification task. In the case presented here, the encoder is trained for the sole task of classification. So if it accumulates somewhere, this is only explainable with respect to the classification task.
> > Note that KMEx prototypes might reveal that some of these accumulation points are due to undesirable artifacts (e.g., Clever Hans). However, since the training of the encoder is plain, fine-tuning it does not involve extra constraints related to the prototypes.
> >
> > We hope that this new formulation removes any ambiguity about the purpose of KMEx and its positioning as an explainability method.

---

> ### Author Response · Authors · 2024-02-22
> **Response to Reviewer J8fa Part 4**
>
> **3. Method section needs more details**
>
> > - In Fig 1, what is D_tsp? It seems like the explanation comes in Section 4 which is way far from Fig. 1.
> > - Also, based on Section 4, with KMEx, D_tsp = 0.0. In that case, 1-D_tsp = 1 for all? But why are the sizes of squares different?
> > - Do we use the same "k" values per class?
> > - What is z? What is p_k? What is epsilon?
> > - It would be always good to define the notation first before using them in the equation.
>
> We would like to thank the reviewer for reporting these regrettable inconsistencies.
> We have edited the caption of Figure 1 to not refer to $D_{tsp}$ so early in the text. The size of the square depends on how many data points activate the prototypes, not $D_{tsp}$. For clarity, we removed this indication from the caption as it is mostly aesthetic and may cause more confusion than it is important to understand the functioning of KMEx.
>
> Following the reviewer's advice, we moved Section 4.2.1 Notations earlier in the text to avoid using objects or notations before they are introduced. To answer more precisely, $z$ is a point in the embedding space $R^D$ where $D$ is an integer indicating the dimension of the embedding space. As for $p_k$, it is a prototype. The notation for the prototypes $p_k$ was unclear. Hence, we now double the indexation $p_{kh}$: $k$ indicate the class and $h$ the index of the prototype in that class.
>
> We also rewrote part of Section 3.1, introducing KMEx to make clear that k-means is not necessarily computed with the same number of prototypes per class. Yet, for simplicity as well as equitable comparison with previous works, this number is fixed throughout our experiments.
>
> > - It seems like the authors claim that trustworthiness of the explanation comes from the black-box model (this part: ...to both, thus similar explanations...). However we cannot say that the black-box model's embedding space is trustworthy.
>
> Trustworthiness, as we introduce it, following ProtoVAE, has two aspects: faithfulness and robustness, both in terms of performance and explanations.
> The idea of faithfulness is that an SEM used as an interpretable alternative to a black-box model should perform on par (accuracy) and yield similar local explanations. The rationale is that if any of these differ too much, then the models are not comparable. As for the robustness, the black-box model serves as a reference in the evaluation.
>
> The computation of local explanation, or saliency maps, involves backpropagating some message over the network, the majority of it consisting of the encoder. Hence, the faithfulness of the explanations is guaranteed by the re-use of the encoder.
> The same applies to the robustness of the explanations.
> In the updated version of the manuscript, we moved away from AI and AD scores and reported the area under the Relevance Order curve (AUROC) instead. This measure proves to better capture the reaction of the models to the gradual masking of the input.
> The importance of the encoder is again validated in Table 4 and Figure 2, where ResNet34 and R34+KMEx report similar curves and thus AUROC.
>
> **3. Experiments (until Section 4.1)**
>
> > - Table 3. Could you explain why some baselines have better results than the proposed method and even baseline method? Can this be the disadvantage of the proposed method?
>
> We agree with the reviewer that some baselines are performing better for different evaluations, especially, in terms of accuracy (Table 3), and robustness of explanations (Table 4). However, we would like to draw the reviewer's attention to the newly added radar plots for all datasets, depicted in Figure 7 in the Appendix. Note that the comparison of SEMs in terms of accuracy essentially comes down to the difference in accuracy relative to the corresponding black-box model. This is depicted in the radar plots by Faithfulness of Accuracy. As can be observed, while a few baselines sometimes outperform the black-box, KMEx still performs on par for all datasets. Further, thanks to the new changes (similarity score and AUROC), the robustness of KMEx has been improved, as can be visualized in this Figure. However, the performance of KMEx in terms of diversity, transparency, and faithfulness of explanations is unmatched by other baselines. Further, in addition to performing on par with other baselines, the advantage of KMEx lies in its simplicity and efficiency in converting a pre-trained black-box into a SEM. This is further supported by the newly added comparison of runtimes of KMEx with other baselines in Tables 11 and 12 in the Appendix.

---

> ### Author Response · Authors · 2024-02-22
> **Response to Reviewer J8fa Part 5**
>
> > - Figure 2. Could we show the qualitative results of other methods that we can compare with? Also good to add how we can interpret PRP (like what is the meaning of the red part)
>
> The interpretation of the PRP maps is available in the caption:
> "PRP maps demonstrating the regions activated by closest prototypes for the test images, exhibiting local explainability."
> This means that according to the PRP rules, the red areas are the regions that are causing the network to position that image close to the prototype in the embedding. Note that we use PRP following previous works, such as ProtoVAE. However, in practice, any other local explanation method could be used. Comparing different local explanation methods, however, is out of the scope of this work.
>
> > - Table 4. It seems that the performance of KMEx is significantly and consistently worse than ProtoPNet. So, in that case, could we understand that KMEx is less robust? In that case, could we use this method for "safety-critical" scenarios where robustness is key?
>
> Following recent works [6], we decided to measure the robustness of the explanations based on the area under the RO curve (AUROC) computed based on probability.
> Previously, we relied on the AI and AD scores related to the similarity with the closest prototypes when the 50\% least important pixels are masked. However, these similarities can not be compared between models as they are not invariant to the scale and spread of the embedding. The use of probabilities solves that issue.
>
> The area under the RO curve also better captures the robustness of both the accuracy and explanations. If the saliency map is robust (the important pixels have the largest values), then the probability of the predicted class remains high for longer as we mask an increasing share of the less important pixels. In the worst case, the AUROC is below 1/(number of classes).
>
> To reply specifically to the poor robustness of R34+KMEx, it should be first noted that, on average, over all the datasets, it achieves better scores than any other SEM tested.
> Nevertheless, the robustness of R34+KMEx is tightly related to that of its encoder.
> Furthermore, specific training techniques can be used to ensure the level of robustness of the "safety-critical scenario" at hand. Given that the encoder is trained for a single loss function, the influence of these regularizations can be seamlessly backtracked.
> We hypothesize that involving a masking scheme during training or fine-tuning might improve the robustness score (whatever the score and model) [7]. Note that this is just an intuition: we did not run any tests.
>
> [6] Schulz, Karl, et al. "Restricting the flow: Information bottlenecks for attribution." arXiv preprint arXiv:2001.00396, 2020.
>
> [7] Kim, Beomsu, et al. "Why are saliency maps noisy? cause of and solution to noisy saliency maps." 2019 IEEE/CVF International Conference on Computer Vision Workshop (ICCVW). IEEE, 2019.
>
>  **4. Experiments (Section 4.2)**
>
> > - Notation part should come much earlier (like Section 3)- Good to define D as well.
>     Good to provide the example of "s".
>
> We have now moved and updated the notation section to Sec. 3.1.
>
> > - Equation (1). It seems like Equation (1) is wrong. See the inside of sigma. We can take out 1/KN part and now we can focus on the nominator with the sigma. As we know, each sample has one nearest prototype. Thus, the sum of the nominator should be N. In that case, D_tsp = 1/K regardless of the method. I do not think this is the definition that the authors want to define.
>
> Indeed the formula was faulty. We apologize for this unfortunate mistake. What we want to quantify is how many prototypes are activated, i.e., that are the closest to at least one data point. The correct formula to estimate this is:
> $$ 1-\frac{1}{KL} \sum_{k=1}^K\sum_{l=1}^L  [[ \exists i \in [1 ... N]: argmax_{\{1 \leq u \leq K\},\{1 \leq v \leq L\}} \Big( \mathfrak{s}(z_i,p_{uv}) = (k,l)\Big) ]],$$
> where $[[ \cdot ]]$ returns 1 if the contained statement is true and 0 otherwise. We update the manuscript accordingly.
>
> > - Also, I think using "testing set" instead of "training set" to evaluate D_tsp would be more meaningful because we would like to use this XAI in a test scenario. Working well with training data may not impact a lot.
>
> We followed the reviewer's advice and recomputed all the Tables using the testing set.

---

> ### Author Response · Authors · 2024-02-23
> **Response to Reviewer J8fa Part 6**
>
> > - What is p. in Equation (4)?
>
> The notation is indeed not the best.
> We changed the notation from prototypes and now use a double indexation: the first for the class and the second for the index of prototype in the class. Accordingly, Equation (4) becomes:
>
> $D_{dvs} = \frac{1}{KL {\log( KL )}}\sum_{k=1}^K\sum_{l=1}^L \mathbf{H}\left( \operatorname{Softmax} \left( {\langle \mathfrak{s}(p_{kl},p_{uv}) \rangle}_{ \substack{{1\leq u \leq K},1\leq v \leq L}} \right)  \right)$
>
> where $K$ is the number of classes, $L$ the number of prototypes per class (here assumed constant).
>
> > - In Table 7, ProtoVAE is much better than KMEx. Is this the right understanding?
>
> The original Tables 7 and 8 were the most affected by the "approximation" made by PyTorch's cdist function. We have now computed the distance d as follows, which trades efficiency for the accuracy of the results:
>
> `` torch.sqrt(torch.square(ebd[:,None,:] - prt[None,:,:]).sum(-1)) ``
>
> In the updated version of Table 7, the advantage of KMEx is clear. On the other hand, the advantage of using KMEx's prototypes on ProtoVAE and ProtoPNet embedding is not that clear anymore. For FLINT, the claim remains. Hence, we were able to keep the last experiment connecting diversity and fairness.

---

### Review · Reviewer_grUW · 2024-02-21

**Summary Of Contributions:**

This paper introduces KMEx (K-Means Explainer). The general idea of this method is to take a base model (e.g., image encoder), learn a set of class prototypes and then predict based on choosing closest prototype for the image. They authors evaluate on several image datasets including MNIST, fMNIST, CIFAR-10 and others. They find roughly find equal performance for the base model (there's a bit delta on a few datasets, but not much). They additionally find the explanations are diverse and high quality.

**Audience:**

Yes

**Broader Impact Concerns:**

No broader impact concerns.

**Claims And Evidence:**

Yes

**Requested Changes:**

While the paper is interesting, the presentation of this paper in its current form needs a bit of work, as the method is hard to follow. For example:

1. In section 3.1, the method is introduced. Yet, this notation is never defined for readers. What is s, z, and p_k?
2. Only scrolling down to 4.2.1 do we see the definitions of notation. Why is the notation introduced in the second subsection of evaluation? This makes it extremely hard to follow.
3. For the similarity measure, why use this measure? Why introduce \epsilon? This needs to be made clearer when introducing the method.

Though the evaluation is comprehensive overall, it would be interesting to see evaluation on more challenging or less data rich situations, because most of the evaluation is currently on datasets that are fairly easy. Are there scenarios where this method breaks down and the prototypes explanations fail to yield useful results? I could imagine this happening in situations with less finetuning data, where the model fails to generalize well to certain test instance.

**Strengths And Weaknesses:**

Pros:
- Simple, easy to use method
- Good results on standard datasets, particularly nice accuracy results for keeping performance similar to non-interpretable method

Cons:
- Presentation is pretty hard to follow in current form. Please move notation definitions out of evaluation section and before you introduce the method. Readers have to hunt around the paper right now to understand it.
- Results mostly on easy datasets, unclear how well the method generalizes to practical settings, though the initial investigation presented here is satisfactory overall

---

> ### Author Response · Authors · 2024-02-23
> **Response to Reviewer grUW**
>
> We thank the reviewer for recognizing the simplicity and efficiency of the proposed model in terms of both the prediction accuracy as compared to the non-interpretable model as well as the diversity of explanations produced. We are happy to answer the reviewer's comments below.
>
> However, before answering the comments, we would like to inform the reviewer of two major changes.
>
> 1. We discovered that PyTorch's ``cdist`` function, in its default setting, computes an approximation of the true distance between two vectors. These undesired approximations have slight repercussions on the scores that we compute except for the accuracy, as the ranking of the similarity scores on which the prediction is based is still preserved. We have therefore re-computed the scores on the previously trained models using the following explicit formula: `` torch.sqrt(torch.square(ebd[:,None,:] - prt[None,:,:]).sum(-1)) ``.
> While results differ slightly, the essence of the message does not change: KMEx offers a quick and efficient way to explain an opaque classifier using prototypes.
>
>
> 2. Following the ablation study suggested by reviewer GmHw, we changed the similarity measure used in KMEx to the negative $\ell_2$ distance. This change as well does not alter the essence of the contribution, yet it makes our model more robust and stable. This is discussed in the \emph{Ablation on similarity measure} section of this answer, as well as added to the Appendix in Sec. A.6.1 of the revised manuscript. In addition to the change in similarity score, we now decided to report the area under the Relevance Ordering (RO) curve with the least relevant pixels removed first in Table 4, instead of AI and AD scores.
> As the RO test is based on class probabilities, it is robust to the scaling and shape of the embedding, thus making it a better choice for comparing the different models. The detailed results are discussed in Sec. 4.2.2 of the main text.
>
> The above changes have led to changes in Tables 5, 6, and 7. Nonetheless, again, the main idea of KMEx remains intact and has, in fact, become more robust. We have further added the missing results for FLINT in Table 6. As observed in Table 6, with the fixed distance implementation, R34+KMEx returns, as expected, the lowest scores for the faithfulness of the explanations.
> The corrected Table 7 highlights the clear advantage of KMEx in terms of computed diversity.
> This advantage can also be seen in Table 8 where for FLINT's embeddings more diverse prototypes are learned using KMEx. However the same is not reflected for ProtoPNet or ProtoVAE's embeddings.
>
>
>
> ### **Presentation is pretty hard to follow in current form**
> We agree that the presentation was a little hard to follow in the original submission. We have now updated the manuscript and moved the notations to Sec. 3.1, which is at the start of the methodology. We hope this change removes any ambiguity regarding the meaning of the various variables used and that it eases the reading of the subsequent sections.
>
> ### **Results mostly on easy datasets, unclear how well the method generalizes to practical settings**
> Our paper intends to present a novel method that can be used as a baseline to compare other SEM, namely KMEx. Since we focused on image classification, we selected *seven* "classic" datasets each with its particularity to illustrate the behavior of our method.
> Despite the relative "easiness" of the *seven* datasets, we can see in Figure 7 of the appendix where we detail the performance of the models on each dataset that often several baselines do not match the accuracy of the vanilla black-box classifier.
>
> Additionally, in Section A.7 of the supplementary, we have reported preliminary results of KMEx on the way more complex CUB200 (already present in the original submission). The classification performance achieved by KMEx is 78.4\% which is again on par with the non-interpretable ResNet34 model with 78.6\% accuracy. This shows the easy scalability of the approach to a comparatively complex dataset with considerable overlap between classes in the embeddings. Additionally, we report initial exploration results for a patch-based KMEx, achieving an accuracy of 70\%. The patches learned by KMEx are shown in Figure 10 in the Appendix. This again confirms the versatility of KMEx.

---

### Decision · Action_Editor_eejj · 2024-04-29

**Recommendation:** Accept with minor revision

**Comment:**

This submission proposes to use a prototype classifier over a learned embedding model to visually attribute the classification decisions of a machine learning system to prototypes. The proposed method is simpler than prior approaches to tie visualizations to decisions (self-explainable models; SEM) and is evaluated similarly to prior work in the subfield.

The concerns of one reviewer, who noted a lack of Evidence for Claims, were on impact ("do not think this can be used in safety-critical scenarios") and on subjective weighing of trade-offs ("[simplicity] does not compensate for ... performance loss") which are not within the scope of the [evaluation criteria of TMLR](https://jmlr.org/tmlr/acceptance-criteria.html). Another reviewer requested more systematic investigations of the clustering step and evaluations on larger datasets; these extensions would be interesting but are not necessary for the work to have an audience with the present claims.

The AE proposes the following revisions:
- to clarify the relationship between the proposed method and existing work on prototype classifiers, especially those used in few-shot learning: [Snell et al. (2017)](https://arxiv.org/abs/1703.05175); [Pahde et al., 2021](https://arxiv.org/abs/2011.08899);
- to avoid aspirational and broad language (e.g., "we introduce our resource-efficient and universal method"; "... push towards more transparent deep learning-based decision-making via class-prototype-based explanations that are *guaranteed* to be diverse and trustworthy...") when describing the contributions of the work and instead describe the proposed method objectively;
- to add nuance to the "main contribution" claim that the evaluation framework is entirely quantitative, as the interpretability component still relies on the visual interpretation of the prototypes and the interpretation of the similarity of a test point to these prototypes;
- avoid any direct claims about the applicability of the approach to safety-critical scenarios, as the evaluation does not support such claims, and these claims are not central to the work;
- to make further improvements in clarity and readability, as noted, could be improved by several reviewers.

**Audience:**

Yes

**Claims And Evidence:**

Yes

---

> ### Author Response · Authors · 2024-05-28
> **Response to the Action Editor eejj**
>
> We thank the AE for the recommendation. We have incorporated the proposed revisions in the revised manuscript. More specifically:
> * We have revised the manuscript to include a discussion about the relationship between our proposed method and existing work on prototype classifiers, particularly those used in few-shot learning as suggested in Section 2.2.
> * We have removed the broad language used, more specifically in abstract, and in the introduction of Section 3.
> * We acknowledge the AE’s point regarding the interpretability component of our method. We have added the suggested nuance to the main contribution.
> * Our discussion on the potential use of self-explainable models in safety-critical scenarios, in the abstract and introduction (referencing Rudin et al.), is meant as general statements and not as a direct claim of our method’s applicability for such scenarios.
> * We have made several revisions to improve the clarity and readability of the manuscript, as suggested by the reviewers.
>
> We hope that these revisions adequately address the concerns raised and believe that they have improved the quality and clarity of our manuscript.